

# Recovery of the 3-dimensional wind and sonic temperature data from a sonic anemometer physically deformed away from manufacture geometrical settings

Xinhua Zhou[1, 2, 3], Qinghua Yang[1, *], Xiaojie Zhen[4], Yubin Li[5], Guanghua Hao[6], Hui Shen[6],

Tian Gao[2], Yirong Sun[2], Ning Zheng[3, *]

[1] Guangdong Province Key Laboratory for Climate Change and Natural Disaster Studies, School of Atmospheric Sciences, Sun Yat-Sen University, Zhuhai 519082, China

[2] CAS-CSI Joint Laboratory of Research and Development for Monitoring Forest Fluxes of Trace Gases and Isotope Elements, Institute of Applied Ecology, Chinese Academy of Sciences, Shenyang 110016, China

[3] Campbell Scientific Incorporation, Logan, Utah 84321, USA

[4] Beijing Techno Solutions Ltd., Beijing 100089, China

[5] Nanjing University of Information Science and Technology, Nanjing 210044, China

[6] National Marine Environmental Forecasting Center, Beijing 100081, China

*Correspondence to*: Qinghua Yang (yangqh25@mail.sysu.edu.cn) and/or Ning Zheng (ning.zheng@campbellsci.com.cn)

**Abstract.** A sonic anemometer (sonic) reports 3-dimensional wind and sonic temperature ($T_s$) by measuring the time of ultrasonic signals flying along each of its three sonic paths whose geometry of lengths and angles in the sonic coordinate system was precisely determined through production calibrations and was embedded into the sonic's firmware. If the sonic path geometry is deformed, although correctly measuring the time, the sonic continues to use its embedded geometry data for internal computations, resulting in incorrect data. However, if the geometry is re-measured (i.e. recalibrated) to update sonic firmware, the sonic can resume reporting correct data. In some cases, where immediate recalibration is not possible, a deformed sonic can be used because ultrasonic signal-flying time is still correctly measured. For example, transportation of a sonic to Antarctica in 2015 resulted in a geometrically deformed sonic. Immediate deployment was critical, so the deformed sonic had been used until a replacement arrived in 2016. To recover data from this deformed sonic, equations and algorithms were developed and implemented into the post-processing software to recover wind data with/without transducer shadow correction and $T_s$ data with crosswind correction. Using two geometric datasets, production calibration and recalibration, post-processing recovered the wind and $T_s$ data from May 2015 to January 2016. The recovery reduced the difference of 9.60 to 8.93 ℃ between measured and calculated $T_s$ to 0.81 to -0.45 ℃, which is within the expected range due to normal measurement errors. The recovered data were further processed to derive fluxes. Since such data reacquisition is time-consuming and expensive, this data recovery approach is a cost-effective and time-saving option applicable to similar cases. The equation development can be a reference to the studies on related topics.

## 1 Introduction

The three-dimensional (3D) sonic anemometer is commonly used for both micrometeorological research and applied meteorology (Horst et al., 2015). It directly measures boundary-layer flows at high measurement rates (e.g., practically 10 to 50 Hz) and outputs wind speeds expressed in the 3D right-handed orthogonal anemometer coordinate system relative to its



structure frame (see Appendix A, hereafter, referred as 3D coordinate system) and sonic temperature calculated from the speed of sound (Campbell Scientific Inc., 1998). Its outputs are commonly used to estimate the fluxes of momentum and sonic temperature and, when combined with fast-response scalar sensors, the fluxes of $CO_2/H_2O$ and other atmospheric constituents.

It has three pairs of sonic transducers structuring three sonic paths (see Fig. 1), each of which is between paired sonic transducers. The three paths are situated as optimized angles in the 3D coordinate system, forming the geometry of sonic anemometer. This geometry is quantitatively defined by the path lengths and path angles that are precisely-measured during production calibration. A sonic anemometer measures the time of ultrasonic signals flying along each path (hereafter, referred as flying time). In reference to the sonic path length, the flying time is used to calculate the speeds of flow and
sound along the path, which will be detailed in Section 4 as the following: according to the angles of three sonic paths, the speeds from the three paths are expressed in the 3D coordinate system for wind and as sonic temperature for air entropy.

A sonic anemometers has geometry embedded into its firmware for internal data processing, allowing output of 3D wind and sonic temperature. However, if it is geometrically deformed from manufacturer's setting at millimeter-scales, or even smaller, due to an unexpected physical impact in transportation, installation, or other handling, the geometry embedded in the
firmware is not representative to the current geometry of this sonic anemometer any more. As a result, the anemometer cannot output correct wind speeds and sonic temperatures, because any deformation in geometry of sonic anemometer can change the relative spatial relationship among its six sonic transducers. If, due to an impact, any transducer is displaced relative to others, the displacement must change at least one of the sonic path lengths and one of the sonic path angles. Fortunately, if geometrical deformation is the only problem, rather than physical damage to the transducers, the sonic
anemometer still can, according to its working physics (Schotland, 1955), correctly perform its flying-time measurements. Due to the change in a sonic path length, although the flying time can be correctly measured, the speeds of air flow and sound along the path are incorrectly computed because the sonic path length embedded in the firmware does not match the true length at the time when the flying time was measured. As a result, the incorrect speeds along with the change in any sonic path angle might lead all 3D wind speeds as well as sonic temperature outputs to be incorrect. These incorrect outputs
are recoverable as correct data because the flying time was correctly measured and the deformed geometry can be re-measured (i.e. recalibrated) by the manufacturer to which the anemometer can be shipped back with care. However, the equations and algorithms for the recovery had not been documented and practiced, which are needed if a sonic anemometer is found to be geometrically deformed in a remote site where its use has to be continued. In such a site, it could take months, seasons, or even longer for a deformed anemometer to be transported back to the manufacturer for geometry re-
measurements, recalibration, and shipped back to the site. In this case, if the measurements were not continued, a measurement-season or -year could be easily missed.

This study demonstrates data recovery from such a case when a sonic anemometer as a component of IRGASON (Integrated $CO_2/H_2O$ Open-Path Gas Analyzer and 3D Sonic Anemometer, Campbell Scientific Inc. 2010b) was geometrically deformed during transportation to Antarctic Zhongshan Station from China in early 2015 and had to be used as planned without a
chance to be shipped back until its replacement of new one arrived at the site early next year. If the deformed sonic was not used then, one measurement-year would have been missed because the only transportation of R/V Xue Long (i.e. Snow Dragon in English) from China to the Zhongshan Station served a round-trip to the site on an annual basis. More importantly, it is a matter of not only one measurement-year but also the 2015 data in particular that were waited by related projects for collaborations. Therefore, the geometrically-deformed sonic anemometer was used as planned and the 2015 data were
acquired. After its field duty was replaced in early 2016, it was shipped, with protection using a pair of buffer bumpers as a



special care, back to the manufacturer of Campbell Scientific Incorporation in the US for re-measurements of its geometry to update its firmware (i.e. recalibration).

Using the measurements of sonic path lengths and sonic path angles for this sonic anemometer from production calibration in April 2014 before its transportation and from recalibration in March 2016 after the field use in the Zhongshan Station, this study aims to develop and verify the equations and algorithms to recover the 2015 data measured using this geometrically deformed sonic anemometer to data as if measured with the this anemometer after recalibration although actually measured before the recalibration, providing a reference to similar cases and/or related topics.

## 2 Site, instrumentation, and data

The observation site was located in the coastal landfast sea ice area of the Zhongshan Station (69 ° 22′ S and 76 ° 22′ E), East Antarctica (Yang et al., 2016; Yu et al., 2017; Zhao et al., 2017). In this area, as influenced by the unique solar cycles, the climate is characterized by the polar night from late March to mid-July and the polar day from mid-November to January. The polar day and the polar night in particular are inhabitable to human life, but drive atmospheric dynamics in a way of interest to human beings (Valkonen et al., 2008); therefore, this region has attracted scientists to measure its surface heat balance; however, the measurements are not an easy task in financial support, technical infrastructure, and administrative management. As such, only few of studies on such measurements have been conducted in this region (e.g., Vihma et al., 2009; Liu et al., 2017).

Supported by National Science Foundation of China, the project: "*Sea ice/snow surface energy budget of the Antarctic Prydz Bay*" was initiated to measure the fluxes of $CO_2/H_2O$, heat, radiation, and atmospheric variables so that the sea ice/snow surface energy budget during both melting and frozen periods can be quantified. For these measurements, the project established two open-path eddy-covariance (OPEC) flux stations in May 2015. One station (see Fig. 2) was configured with IRGASON (SN: 1131) for the fluxes of $CO_2/H_2O$, sensible heat, and momentum; one 4-way net radiometer (model: CNR4, Kipp & Zonen, Delft, The Netherlands) for net radiation from incoming short-wave, outgoing short-wave, incoming long-wave, and outgoing long-wave radiation components; one temperature and relative humidity probe (model: HMP155A, SN: H5140031, Vaisala, Helsinki, Finland) inside a 14-plate naturally-aspirated radiation shield of model 41005 for air temperature and air relative humidity; and one infrared radiometer (model: SI-111, SN: 2962, Apogee, UT, USA) for surface temperature. Later 2015, a CSAT3B (Campbell Scientific Inc. UT, USA) was added for additional data of 3D wind and sonic temperature. This OPEC station is also equipped with a built-in barometer (Model: MPXAZ6115A, Freescale Semiconductor, TX, USA) for atmospheric pressure and a built-in 107 temperature probe (Model: 100K6A1A, BetaTherm, Finland) inside a 6-plate naturally-aspirated radiation shield of model 41303-5A for air temperature, the IRGASON was connected to and controlled by an EC100 electronic module (SN: 1542, OS: Rev 04) that, in turn, was connected to and instructed by a central CR3000 Measurement and Control Datalogger (SN: 7720, OS 25) for these sensor measurements, data processing, and data output. While receiving the data output from EC100 at 10 Hz, the CR3000 also controlled and measured slow response sensors at 0.1 Hz such as the CNR4, HMP155A, and others in support to this study. EasyFlux_CR3OP (version 1.00, Campbell Scientific Inc. 2016) was used inside CR3000. The data of 3D wind, sonic temperature, $CO_2/H_2O$ amount, atmospheric pressure, diagnosis codes for the 3D sonic anemometer and open-path infrared gas analyzer, air temperature, and relative humidity were stored 10 records per second (i.e., 10 Hz). The data from all sensors were computed and stored by the CR3000 every half-hour interval.

## 3 Data check and instrument diagnosis


Immediately after the station started to run, all measured values were checked. Unfortunately, the sonic temperature from 3D sonic anemometer was found to be incorrect because it was around 10 ℃ higher than the air temperature from HMP155A or 100K6A1A. Given $H_2O$ density about 1.00 g m$^{-3}$ and air temperature about -20 ℃ then, sonic temperature should be around 0.13 ℃ higher than air temperature [see Eq. (5) in Schotanus et al., (1983)] if the sonic temperature was measured, although

impossible, without an error. Further diagnosis for sonic anemometer measurements found that the sonic temperature values from the three sonic paths were unexpectedly various individually around -12, 5, and -7 ℃ as shown by Device Configuration (Campbell Scientific Inc. UT, USA) connected to EC100 through a notebook computer while the station was running. Apparently, the largest absolute difference in sonic temperature among the three paths reached 17 ℃ although this difference from an IRGASON sonic anemometer was expected < 1 ℃. Such a large unexpected absolute difference (e.g.

17 ℃) among the three values from the three sonic paths might be caused by the geometrical deformation of sonic anemometer. To confirm the diagnosis, the body of IRGASON was visually examined and painting on the knuckle of side one (i.e., 1$^{st}$ sonic path) among the top three claws was found off as apparently impacted (see Fig. 3). Therefore, with confidence, it was concluded that the incorrect outputs of sonic temperature were caused by the geometrical deformation of sonic anemometer while being transported to Antarctica from China. The deformation also might cause the incorrect outputs

of 3D wind. Therefore, this IRGASON should have been shipped back to manufacturer for re-measurements of its geometry to update its OS (recalibration). However, as addressed in Introduction, the 2015 data would have been missed if it were shipped back to manufacturer. To make measurements as planned, this IRGASON continued its field duty until next round-trip of R/V Xue Long to Antarctica from China by the end of 2015 when its replacement from the manufacturer arrived at the site.

In early 2016, it was replaced in the field and was shipped back to the manufacturer where it was re-measured for sonic geometry in recalibration process on March. The re-measurements verified our diagnosis conclusion that the IRGASON sonic anemometer was geometrically deformed (see Table A1 in Appendix A). Therefore, the 2015 data from this sonic anemometer need to be recovered as if measured by the same anemometer after recalibration although these data were acquired from the measurements before the recalibration.

**4 Algorithm to recover the data of 3D wind and sonic temperature**

An IRGASON sonic anemometer measures wind flows along its three non-orthogonal sonic paths (i.e. the three sonic paths non-orthogonally situated each other, see Fig. 1), each of which is between a pair of sonic transducers. Sensing each other in each sonic path, the pair separately pulse two ultrasonic signals in opposite directions at the same time. The signal pulsed by the transducer facing to air flow direction along the sonic path takes less time to be sensed by its paired one than the one

pulsed by the transducer against the air flow direction. In a path, the flying time of ultrasonic signal upward [$t_{ui}$ where subscript $i$ can be 1, 2, or 3, denoting the sequential order of sonic path (see Fig. 1). This subscript denotes the same throughout] and downward ($t_{di}$) are measured by the sonic anemometer (Campbell Scientific Inc. 1998). In the case as shown in Fig. 1 for the 3$^{rd}$ sonic path, or $i = 3$, the flying time of ultrasonic signal in the path upward is given by:

$$t_{u3} = \frac{d_3}{c_3 + u_3} \qquad (1)$$

where, along the 3$^{rd}$ sonic path, $d_3$ is its length precisely measured during production or recalibration process using a Coordinate Measurement Machine (CMM), $c_3$ is the speed of sound, and $u_3$ is the speed of air flow (see Fig. 1); and the flying time of ultrasonic signal downward is given by:





$$t_{d3} = \frac{d_3}{c_3 - u_3} \tag{2}$$

### 4.1 Recover 3D wind data

### 4.1.1 Algorithm of sonic anemometer to output the 3D wind data

Equations (1) and (2) lead to:

$$u_3 = \frac{d_3}{2}\left[\frac{1}{t_{u3}} - \frac{1}{t_{d3}}\right] \tag{3}$$

Using the same procedure, $u_1$ and $u_2$ (see Fig. 1) can be derived as the same form. In reference to Eq. (3), equation for $u_i$; where subscript $i = 1$, 2, or 3; can be expressed as

$$u_i = \frac{d_i}{2}\left[\frac{1}{t_{ui}} - \frac{1}{t_{di}}\right] \tag{4}$$

Similar to $d_3$, $d_1$ and $d_2$ are also precisely measured using CMM. The three flow speeds of $u_i$ ($i = 1$, 2, or 3) measured from the three non-orthogonal paths are then expressed in the 3D right-handed orthogonal instrument coordinate system of $x$, $y$, and $z$; where $x$ and $y$ are the horizontal coordinate axes and $z$ is the vertical axis; through a transform matrix $\mathbf{A}$ as the 3D wind speeds ($u_x$, $u_y$, and $u_z$) commonly used in practices:

$$\begin{bmatrix} u_x \\ u_y \\ u_z \end{bmatrix} = \mathbf{A} \begin{bmatrix} u_1 \\ u_2 \\ u_3 \end{bmatrix} \tag{5}$$

where the 3D right-handed orthogonal instrument coordinate system (hereafter, sonic coordinate system. see Figs. 1 and A1) is defined by its origin at the center of sonic measurement volume, the $u_x$-$u_y$ plain parallel to the imagery plain leveled by a built-in bulb in the anemometer structure, and the $u_y$-$u_z$ plain through the 1[st] sonic path and $\mathbf{A}$ is a 3×3 matrix constructed using precisely measured geometry of the sonic paths in angles relative to the 3D coordinate system (see its derivations in Appendix A). Matrix $\mathbf{A}$ is unique for each sonic anemometer and is embedded in its firmware; therefore, the 3D wind data

outputted from the anemometer are the three components of $u_x$, $u_y$ and $u_z$ in the 3D coordinate system.

Due to shadowing from the sonic transducer itself (transducer shadowing), the measured $u_i$ is assumed to be lower than its true value in magnitude (Wyngaard and Zhang, 1985; Kaimal and Finnigan, 1994). As denoted by $u_{Ti\_n}$ where subscript $T$ indicates "True" and subscript $\_n$ indicates that $u_{Ti\_n}$ was estimated from $n$ counts of iterations of transducer shadow correction as shown in Appendix B, this true value is assumed to be approached through the transducer shadow correction

from $u_i$. Now, the shadow correction was implemented as an option if IRGASON OS 5 or newer. Therefore, based on the option, Eq. (5) alternatively can be expressed as:

$$\begin{bmatrix} u_x \\ u_y \\ u_z \end{bmatrix} = \mathbf{A} \begin{bmatrix} u_{T1\_n} \\ u_{T2\_n} \\ u_{T3\_n} \end{bmatrix} \tag{6}$$

According to Wyngaard and Zhang (1985), the correction equation for the sonic transducer size and sonic path geometry of IRGASON sonic anemometer is given by:




$$u_{Ti\_1} = \frac{u_i}{0.84 + 0.16 \sin \alpha_i} \qquad (7)$$

where $\alpha_i$ is the angle of the total wind vector to the wind vector along sonic path $i$ and is unknown before the two vectors are accurately estimated, but, referencing Figs. 1 and 4, the $\sin\alpha_i$ in Eq. (7) can be alternatively expressed as a function of flow speed values to lead Eq. (7) as

$$u_{Ti} = \frac{u_i}{0.84 + 0.16 \dfrac{\sqrt{U_T^2 - u_{Ti}^2}}{U_T^2}} \qquad (8)$$

where $U_T$ is the magnitude of total true wind vector, given by

$$U_T = \sqrt{u_x^2 + u_y^2 + u_z^2} \qquad (9)$$

In Eq. (8), all independent variables are actually related to the variables in Eq. (5). As such, using this equation, $u_{Ti}$ can be computed; however, there are two inconvenient issues in this equation application to transducer shadow corrections: 1) an

analytical solution for $u_{Ti}$ is not easily available because $u_{Ti}$ is in a 2$^{nd}$ order term under a square root in the right hand of Eq. (8) although $u_{Ti}$ is analytically expressed in its left hand side and 2) $U_T$ is not available either because $u_x$, $u_y$, and $u_z$ are derived from $u_1$, $u_2$, and $u_3$ before the transducer shadow corrections. Fortunately, the corrections are small in magnitude as shown in Eq. (8); therefore, $u_i$ is closed to $u_{Ti}$. As a result, $u_x$, $u_y$, and $u_z$ from Eq. (5) are close to those from Eq. (6). Accordingly, iteration algorithm may be a right approach to the corrections using Eq. (8), or to estimation of $u_{Ti}$.

For the 1$^{st}$ iteration, $u_{Ti}$ in the right hand of Eq. (8) could be replaced with $u_i$ as its estimation. Given that $U_T$ should be calculated using $u_x$, $u_y$, and $u_z$ from Eq. (6), before the shadow corrections, $U_T$ can be estimated using $u_x$, $u_y$, and $u_z$ from Eq. (5). See Appendix B: Iteration algorithm for sonic transducer shadow corrections. The iterations ensure that the difference in $u_x$, $u_y$, or $u_z$ between last and previous iterations are $< 1 \, \text{mm s}^{-1} \approx 1.96\sigma < 1$ where $\sigma$ is the maximum precision (i.e. standard deviation at constant wind) among $u_x$, $u_y$, and $u_z$ (Campbell Scientific Inc., 2010b). The $u_{T1\_n}$, $u_{T2\_n}$, and $u_{T3\_n}$ from

the last interaction are finally used for Eq. (6) to compute the 3D wind of $u_x$, $u_y$, and $u_z$ as sonic anemometer output.

### 4.1.2 Procedure to recover 3D wind data

As addressed in Eqs. (4) to (6), a sonic anemometer measures $t_{ui}$ and $t_{di}$ to calculate the 3D wind of $u_x$, $u_y$, and $u_z$; therefore, sonic path lengths ($d_i$) in Eq. (4) and transform matrix $\mathbf{A}$ in Eqs. (5) and (6) are embedded into the firmware of sonic anemometer in manufacture processes. If the anemometer was physically deformed in transportation, installation, or other

handling; the sonic path lengths and sonic path angles must be changed from what they were at the time when $d_i$ and $\mathbf{A}$ were embedded into its firmware; therefore, $d_i$ in Eq. (4) and sonic path angles reflected by $\mathbf{A}$ in Eqs. (5) and (6) are no longer valid for this anemometer. Consequently; the output of $u_x$, $u_y$, and $u_z$ still based on embedded $d_i$ and $\mathbf{A}$ from production or calibration process are erroneous. To correct the erroneous output; $u_x$, $u_y$, and $u_z$ need to be transformed back to $t_{ui}$ and $t_{di}$ and to be recalculated using $t_{ui}$ and $t_{di}$ based on the true sonic path lengths and true sonic path angles at the time when $t_{ui}$ and $t_{di}$

were measured in the field by the sonic anemometer physically deformed away from manufacture geometrical settings before its field deployment.

For the true sonic path lengths and true sonic path angles, IRGASON (SN: 1131) was returned to the manufacturer in the way as described in Section 3. In the same way as in the manufacture process, the lengths and angles were re-measured using CMM. The re-measured lengths are denoted by $d_{Ti}$ ($i$ = 1, 2, or 3) and the re-measured angles were used to reconstruct the

transform matrix A as $\mathbf{A}_T$ (see **Appendix A**). Both $d_{Ti}$ and $\mathbf{A}_T$ are used to update the firmware of this IRGASON for future



field uses and to correct $u_x$, $u_y$, $u_z$ and $T_s$ (sonic temperature, see Section. 4.2) that were outputted in the field before the re-measurements. The correction procedures are different for the output of $u_x$, $u_y$, $u_z$ with or without transducer shadow corrections.

**i. With transducer shadow corrections**

Transfer $u_x$, $u_y$, and $u_z$ in the 3D coordinate system to the flow speeds along the sonic paths after transducer shadow corrections.

$$\begin{bmatrix} u_{T1\_n} \\ u_{T2\_n} \\ u_{T3\_n} \end{bmatrix} = \mathbf{A}^{-1} \begin{bmatrix} u_x \\ u_y \\ u_z \end{bmatrix} \tag{10}$$

Using Eq. (B5), flow speed along the $i^{th}$ sonic path before transducer correction ($u_i$) can be expressed as

$$u_i = u_{Ti\_n}\left(0.84 + 0.16\frac{\sqrt{U_T^2 - u_{Ti\_m}^2}}{U_T}\right) \tag{11}$$

where $U_T$ can be calculated using Eq. (9) and $u_{Ti\_m}$ can be reasonably approximated using $u_{Ti\_n}$ because $u_{Ti\_m}$ and $u_{Ti\_n}$ are close enough to ensure $u_x$, $u_y$, and $u_z$ to converge at their measurement precisions (see Appendix B). Using $u_i$ and $d_i$, the time term inside the square bracket in Eq. (4) can be recovered

$$\left[\frac{1}{t_{ui}} - \frac{1}{t_{di}}\right] = \frac{2u_i}{d_i} \tag{12}$$

Also according to Eq. (4) and using $d_{Ti}$, the speed of air flow along the $i^{th}$ sonic path can be recalculated as $u_{ci}$:

$$u_{ci} = \frac{d_{Ti}}{2}\left[\frac{1}{t_{ui}} - \frac{1}{t_{di}}\right] \tag{13}$$

Further replacing $u_i$ with $u_{ci}$ in the iteration algorithm for sonic transducer shadow corrections in Appendix B, $u_{ci}$ is corrected for transducer shadowing as $u_{cTi\_n}$. Using Eq.(6), the recovered vector of 3D wind in the 3D coordinate system $\begin{bmatrix} u_{cx} & u_{cy} & u_{cz} \end{bmatrix}'$ can be expressed as:

$$\begin{bmatrix} u_{cx} \\ u_{cy} \\ u_{cz} \end{bmatrix} = \mathbf{A}_T \begin{bmatrix} u_{cT1\_n} \\ u_{cT2\_n} \\ u_{cT3\_n} \end{bmatrix} \tag{14}$$

**ii. Without transducer shadow corrections**

Transfer $u_x$, $u_y$, and $u_z$ in the 3D coordinate system to the flow speeds along individual sonic paths

$$\begin{bmatrix} u_1 \\ u_2 \\ u_3 \end{bmatrix} = \mathbf{A}^{-1} \begin{bmatrix} u_x \\ u_y \\ u_z \end{bmatrix} \tag{15}$$

Using Eqs. (12) and (13), the speed of flow along the $i^{th}$ sonic path ($u_{ci}$) is recalculated (i.e. recovered). Based on Eq. (5), the recovered speeds of flow along the three sonic paths can be expressed in the 3D coordinate system as



$$\begin{bmatrix} u_{cx} \\ u_{cy} \\ u_{cz} \end{bmatrix} = \mathbf{A}_T \begin{bmatrix} u_{c1} \\ u_{c2} \\ u_{c3} \end{bmatrix} \tag{16}$$

### 4.2 Recover sonic temperature data

#### 4.2.1 Algorithm of sonic anemometer to output sonic temperature

Equations (1) and (2) also lead to:

$$c_3 = \frac{d_3}{2} \left[ \frac{1}{t_{u3}} + \frac{1}{t_{d3}} \right] \tag{17}$$

Using the same procedure, $c_1$ and $c_2$ (see Figs. 1 and 5) can be derived as the same form. In reference to Eq. (17), equation for $c_i$; where subscript $i = 1, 2,$ or 3; can be expressed as

$$c_i = \frac{d_i}{2} \left[ \frac{1}{t_{ui}} + \frac{1}{t_{di}} \right] \tag{18}$$

Here, $c_i$ is the measured speed of sound along the sonic path $i$ (see Fig. 5). When the crosswind ($u_{\perp i}$), or wind normal to the sonic path $i$, is zero; $c_i$ is the true speed of sound ($c_{Ti}$). Unfortunately, crosswind rarely is zero and $c_i$ needs to be corrected to $c_{Ti}$. According to Figs. 1 and 5, the true speed of sound is given by:

$$c_{Ti} = \frac{c_i}{\cos \alpha_i} = \frac{c_i}{c_i / \sqrt{c_i^2 + u_{\perp i}^2}} = \sqrt{c_i^2 + u_{\perp i}^2} \tag{19}$$

Referencing the diagram for wind vectors in the left side of Fig. 5, this equation can be expressed as

$$c_{Ti}^2 = c_i^2 + U_T^2 - u_{Ti}^2 \tag{20}$$

According to the definition of sonic temperature (Kaimal and Finnigan, 1994), the sonic temperature (K) along the $i^{th}$ sonic path ($T_{si}$) should be expressed as:

$$T_{si} = \frac{c_{Ti}^2}{\gamma_d R_d} \tag{21}$$

where $\gamma_d$ (1.4003) is the ratio of dry air specific heat at constant pressure (1,004 J K$^{-1}$ kg$^{-1}$) to dry air specific heat at constant
volume (717 J K$^{-1}$ kg$^{-1}$) and $R_d$ (287.04 J K$^{-1}$ kg$^{-1}$) is gas constant for dry air. The sonic temperature outputted from sonic anemometer ($T_s$ in °C) is the average from the three sonic paths, given by:

$$T_s = \frac{1}{3} \sum_{i=1}^{3} T_{si} - 273.15 = \frac{1}{3\gamma_d R_d} \sum_{i=1}^{3} c_{Ti}^2 - 273.15 \tag{22}$$

Substituting $c_{Ti}$ with Eq. (20) and then substituting $c_i$ with Eq. (18), $T_s$ can be expressed as:

$$T_s = \frac{1}{3\gamma_d R_d} \left\{ \sum_{i=1}^{3} \left[ \frac{d_i^2}{4} \left( \frac{1}{t_{ui}} + \frac{1}{t_{di}} \right)^2 - u_{Ti}^2 \right] + 3U_T^2 \right\} - 273.15 \tag{23}$$

#### 4.2.2 Procedure to recover sonic temperature data





Equation (23) indicates that, given $d_i$, a sonic anemometer estimates sonic temperature using its measured flying time of $t_{ui}$ and $t_{di}$, the flow speeds along the sonic paths ($u_i$ or $u_{Ti}$ if corrected for transducer shadowing) that are also calculated from $t_{ui}$ and $t_{di}$ (see Eq. 4), and the resultant wind speed ($U_T$, i.e., the total wind) computed using Eq. (9) inside which the three wind components in the 3D coordinate system are transformed from $u_i$ using $\mathbf{A}$ as explained by Eq. (5) without transducer

corrections or from $u_{Ti}$ also using $\mathbf{A}$ as explained by Eq. (6) with transducer corrections. As discussed in Section 4.1.2, if a sonic anemometer is geometrically deformed in an incident, the sonic path lengths and sonic path angles may be changed from what they were at the time when $d_i$ and $\mathbf{A}$ were embedded into its firmware; therefore, $d_i$ in Eq. (23) and $\mathbf{A}$ in Eqs. (5) and (6) for $u_i/u_{Ti}$ and $U_T$ in Eq. (23) are no longer valid for this sonic anemometer. As a result; its output of $u_x$, $u_y$, $u_z$, and $T_s$ still based on embedded $d_i$ and $\mathbf{A}$ must not be representative the field wind to be measured. In Section of 4.1, the procedure

to recover 3D wind data was developed using re-measured sonic path lengths ($d_{Ti}$) and re-determined sonic path angles for $\mathbf{A}_T$. The procedure to recover sonic temperature data also needs to be developed using $d_{Ti}$ and recovered 3D wind data in this section as follows.

Based on Eq. (20), the recovered speed of sound from sonic path $i$ after crosswind corrections can be expressed as

$$c_{cTi}^2 = c_{ci}^2 + U_{cT}^2 - u_{cTi}^2 \tag{24}$$

where $c_{ci}$ is the recovered speed of sound along sonic path $i$ and $U_{cT} = \sqrt{u_{cx}^2 + u_{cy}^2 + u_{cz}^2}$ . After replacement of $c_{Ti}^2$ with

$c_{cTi}^2$ in Eq. (22), the recovered sonic temperature ($T_{cs}$ in ℃) can be written as:

$$T_{cs} = \frac{1}{3\gamma_d R_d} \sum_{i=1}^{3} c_{cTi}^2 - 273.15 \tag{25}$$

Now, the term of $c_{cTi}^2$ needs to be derived. Subtracting Eq. (20) from (24) leads to:

$$c_{cTi}^2 = c_{Ti}^2 + \left(c_{ci}^2 - c_i^2\right) + \left(U_{cT}^2 - U_T^2\right) - \left(u_{cTi}^2 - u_{Ti}^2\right) \tag{26}$$

Using this equation to substitute $c_{cTi}^2$ in Eq. (25), denoting $U_{cT}^2 - U_T^2$ with $\Delta U_{cT}^2$ and denoting $u_{cTi}^2 - u_{Ti}^2$ with $\Delta u_{cTi}^2$ lead to:

$$T_{cs} = T_s + \frac{1}{3\gamma_d R_d} \sum_{i=1}^{3} \left[\left(c_{ci}^2 - c_i^2\right) + \Delta U_{CT}^2 - \Delta u_{cTi}^2\right] \tag{27}$$

In this equation, the term of $c_{ci}^2 - c_i^2$ is still unknown. Based on Eq. (18), $c_{ci}^2$ is given by:

$$c_{ci}^2 = \frac{d_{Ti}^2}{4}\left[\frac{1}{t_{ui}} + \frac{1}{t_{di}}\right]^2 \tag{28}$$

Accordingly, the unknown term is given by:

$$c_{ci}^2 - c_i^2 = \frac{d_{Ti}^2}{4}\left[\frac{1}{t_{ui}} + \frac{1}{t_{di}}\right]^2 - \frac{d_i^2}{4}\left[\frac{1}{t_{ui}} + \frac{1}{t_{di}}\right]^2$$

$$= \frac{1}{4}\left[\frac{1}{t_{ui}} + \frac{1}{t_{di}}\right]^2 \left(d_{Ti}^2 - d_i^2\right) \tag{29}$$

$$= c_i^2 \frac{\Delta d_{Ti}^2}{d_i^2}$$





In this equation, only unknown variable is $c_i^2$. Based on Eq. (20), this equation can be expressed as:

$$c_{ci}^2 - c_i^2 = \left(c_{Ti}^2 - U_T^2 + u_{Ti}^2\right)\frac{\Delta d_{Ti}^2}{d_i^2} \tag{30}$$

In the right hand side of this equation, $c_{Ti}^2$ is unknown only. However, the whole term in the right hand of Eq. (29) mathematically is a differential term in which $c_{Ti}^2$ can be reasonably approximated using its neighbor value as close as possible to $c_{Ti}^2$. The average of $c_{T1}^2, c_{T2}^2,$ and $c_{T3}^2$ can be calculated from Eq. (22) because $T_s$ is an output variable of sonic anemometer. Without a measurement error and random error, the three $c_{Ti}$ should be the same independent of flow speed because they are the true speed of sound instead of measured speed of sound along an individual sonic path (Schotanus et al., 1983; Liu et al., 2001); Therefore, $c_{Ti}^2$ can be reasonably approximated using the average of three $c_{Ti}^2$ as $c_T^2$, given by:

$$c_{ci}^2 - c_i^2 = \left(c_T^2 - U_T^2 + u_{Ti}^2\right)\frac{\Delta d_{Ti}^2}{d_i^2} \tag{31}$$

where $c_T^2$ can be computed from Eq. (22) as.

$$c_T^2 = \gamma_d R_d \left(T_s + 273.15\right) \tag{32}$$

Due to the replacement of $c_{Ti}^2$ with $c_T^2$, the relative error of whole term in the right hand side of Eq. (31) would be < 4% even if the variability in sonic temperature due to the difference among $c_{Ti}^2$ values reaches 10 ℃ at air temperature of -30 ℃ without wind (i.e., $U_T = 0$ and $u_{Ti} = 0$), which would be the worst case. Substituting the term of $c_{ci}^2 - c_i^2$ in Eq. (27) with Eq. (31) leads to

$$T_{cs} = T_s + \frac{1}{3\gamma_d R_d}\sum_{i=1}^{3}\left[\left(c_T^2 - U_T^2 + u_{Ti}^2\right)\frac{\Delta d_{Ti}^2}{d_i^2} + \Delta U_{cT}^2 - \Delta u_{cTi}^2\right] \tag{33}$$

In the right hand side of this equation, the whole term after $T_s$ is the sonic temperature recovery term interpretable.

**5 Application**

For our case without a transducer shadow correction, Eqs. (15), (12), (13), and (16) were sequentially used to recover the 3D wind data. In a case of transducer shadow correction in option, Eqs. (10) to (16) are used. Based on the data of 3D wind from the recovery process, Eqs. (9), (32), and (33) were used to recover the sonic temperature data. The whole recovery processes large data files (10 records per second), not only using these equations, but also operating the matrixes (A3) to (A5) (see Appendix A) for Eqs. (15) and (16) along with the data of sonic paths lengths in Table A1 for Eqs. (12) and (13). Apparently, the recovery process is a huge work load in computation. As such, these equations, matrixes, and data were implemented into a software package: "Sonic Data Recovery for IRGASON/CSAT3/A/B Used in Geometrical Deformation after Production/Calibration" whose interface is shown in Fig. 6 and Appendix C. As long as the path lengths and matrixes from production/calibration and from recalibration are input into the software as instructed by the interface (see Fig. 6), the software automatically recover the data in batches.

**6 Verification**



In our station, an additional anemometer for wind was not under deployment when this studied IRGASON was used in its deformed state; therefore, no data were available to verify the recovered 3D wind data. However, the algorithms as addressed using Eq. (10) to Eq. (16) to recover the 3D wind data are solid without any estimation and the recovered 3D wind data are not necessary to be verified.

Fortunately, the data to verify sonic temperature are available in this station. Air temperature, relative humidity, and atmospheric pressure were measured using research grade sensors of HMP155A and IRGASON built-in barometer and the data of these variables also stored at 10Hz (10 records per second). These data can be used to estimate the sonic temperature (see Appendix D: Sonic temperature from air temperature, relative humidity, and atmospheric pressure). The recovered data of sonic temperature using Eq. (33) were compared to the calculated sonic temperature over the range of sonic temperature

for three representative values: -20.01 $\pm$ 0.14 ℃ in Fig. 7a, -9.06 $\pm$ 0.13 ℃ in Fig. 7b, and -1.90 $\pm$ 0.22 ℃ in Fig. 7c. The difference between measured (i.e., unrecovered) and calculated sonic temperature values of 9.60 $\pm$ 0.14 ℃ in Fig. 7a, 9.53 $\pm$ 0.17 ℃ in Fig. 7b, and 8.93 $\pm$ 0.24 ℃ in Fig. 7c was narrowed to 0.99 $\pm$ 0.14 ℃, 0.57 $\pm$ 0.17 ℃, and -0.25 $\pm$ 0.24 ℃, respectively, as the difference between recovered and calculated sonic temperature values. Given the accuracy of $\pm$0.5 ℃ in sonic temperature from IRGASON sonic anemometer (Personal communication with Larry Jacobsen who is the designer of

sonic anemometer) and the accuracy of $\pm$0.2 ~ 0.3 ℃ in air temperature below 0 ℃ and 1.2% in relative humidity from HMP155A (Campbell Scientific Inc., 1990), from which the calculated sonic temperature was derived (see Appendix D), recovered sonic temperature data can be reasonably judged as satisfactory if the difference in mean sonic temperature between recovered and calculated ranges within $\pm$0.80 ℃ or even wider that could be considered a likelihood range of possible difference between correctly measured and calculated sonic temperature. As shown in Fig. 7, Eq. (33) apparently

did an excellent job in recovering the sonic temperature data measured using sonic anemometer in its deformed state, but is barely satisfactory in case of Fig. 7a (i.e., 0.99 $\pm$ 0.14 ℃, the difference in sonic temperature between recovered and calculated) although the range of 0.99 $\pm$ 0.14 ℃ is not significantly different from $\pm$0.80 ℃. The bare satisfactory might be caused by the approximation of $c_{Ti}$ from $c_T$ that is fully valid if all $c_{Ti}$ are not measured by a sonic anemometer in its deformed state, but not a case in this study.

According to Eq. (22), it is impossible to have an individually $c_{Ti}$ from $T_s$ which is the sole output for sonic temperature from any sonic anemometer. Now, the average of $c_{T1}^2, c_{T2}^2,$ and $c_{T3}^2$ is known and the changes in sonic path lengths are known. It is possible to estimate the difference among the three speeds of sound and to adjust their average ( $c_T^2$ ) to $c_{T1}^2, c_{T2}^2,$ and $c_{T3}^2$ in approximation although the exact values are impossible. The adjusted values can reflect the variability among $c_{Ti}^2$ at some degree and are reasonably expected to improve the data recovery.

**7 Adjustment**

The measured speed of sound after crosswind correction ($c_{Ti}$) is independent of wind speed (Liu et al. 2001). Given air density and atmospheric pressure (Barrett and Suomi, 1949), without wind, $c_{Ti}$ is equal to the measured speed of sound ($c_i$) from sonic path $i$ [see Eq. (19)]. In this case again without wind, $t_{ui}$ and $t_{di}$ in Eq. (18) are the same and can be denoted by $t_i$. Accordingly, Eq. (18) in this case is equivalent to

$$c_{Ti} \equiv \frac{d_i}{t_i} \qquad\qquad (34)$$





In Eq. (33), $c_T^2$ is the average of three squared $c_{Ti}$ [see Eqs. (22) and (32)], but an individual $c_{Ti}$ is unknown; therefore, for recovery improvement, it has to be estimated from $c_T^2$ through a reasonable adjustment. The difference in magnitude between $c_T^2$ and $c_{Ti}^2$ must be related to the $c_{Ti}^2$ error due to the geometrical deformation of sonic animometer. Squaring both sides of Eq. (34) leads to

$$c_{Ti}^2 = \frac{d_i^2}{t_i^2} \tag{35}$$

The total differentiation of $c_{Ti}^2$ is given by

$$\Delta c_{Ti}^2 = \frac{2d_i}{t_i^2} \Delta d_i - \frac{2d_i^2}{t_i^3} \Delta t_i \tag{36}$$

Given the flying time is correctly measured by a sonic anemometer (i.e., $\Delta t_i = 0$) even in its geometrical deformation, this equation becomes

$$\Delta c_{Ti}^2 = \frac{2d_i}{t_i^2} \Delta d_i = c_{Ti}^2 \frac{2\Delta d_i}{d_i} = c_{Ti}^2 \frac{2(d_i - d_{Ti})}{d_i} \tag{37}$$

Mathematically in differentiation, $c_{Ti}^2$ can be reasonably approximated by $c_T$, given by

$$\Delta c_{Ti}^2 \approx 2c_T^2 \left(1 - \frac{d_{Ti}}{d_i}\right) \tag{38}$$

This is the error of $c_{Ti}^2$ away from $c_T^2$. This error can be reasonably used to represent the deviation of $c_{Ti}^2$ away from $c_T^2$. The deviations of three $c_{Ti}^2$ values away from $c_T^2$ are the measures of variability among three $c_{Ti}^2$ away from $c_T^2$.

15    Although an individual $c_{Ti}^2$ is unknown, the average of three $c_{Ti}^2$ is known as $c_T^2$. This average should be unchanged after adjustments because of the adjustment within the variability among $c_{Ti}^2$ away from $c_T^2$. If the average of adjusted $c_{Ti}^2$ is not equal to $c_T^2$, all adjusted $c_{Ti}^2$ should be added or subtracted with the same constant to make the average of three adjusted $c_{Ti}^2$ values as $c_T^2$, but the variability among $c_{Ti}^2$ values is kept the same. This constant must be the mean of three $\Delta c_{Ti}^2$ values. Based on these analyses, the adjustment of $c_T^2$ to $c_{Ti}^2$ can be constructed as

$$c_{Ti}^2 \equiv c_T^2 + \left(\Delta c_{Ti}^2 - \frac{1}{3}\sum_{i=1}^{3} \Delta c_{Ti}^2\right) \tag{39}$$

Using this equation to replace $c_{Ti}^2$ in Eq. (30) and the resultant equation with this replacement then is used to $c_{ci}^2 - c_i^2$ in Eq. (27) as

$$T_{cs} = T_s + \frac{1}{3\gamma_d R_d} \sum_{i=1}^{3} \left\{ \left[ c_T^2 + \left(\Delta c_{Ti}^2 - \frac{1}{3}\sum_{j=1}^{3} \Delta c_{Tj}^2\right) - U_T^2 + u_{Ti}^2 \right] \frac{\Delta d_{Ti}^2}{d_i^2} + \Delta U_{cT}^2 - \Delta u_{cTi}^2 \right\} \tag{40}$$

In the right hand side of equation, the whole term after $T_s$ is the adjusted sonic temperature recovery term.

25    The data ever recovered using Eq. (33) also were recovered again using Eq. (40). Apparently, this equation did a better job than Eq. (33). The difference in sonic temperature between recovered and calculated was reduced to 0.81 $\pm 0.14$ ℃, 0.38 $\pm$ 0.17 ℃, and -0.45 $\pm 0.24$ ℃, respectively, as shown from panels a to c in Fig. 7. These values for the difference fall into the





range of ±0.80 ℃ in statistical sense. Equation (40) is believed to do a better job than Eq. (33), although, that is satisfactory. Eventually, Eq. (40) was used for data recovery and was incorporated into the software as shown in Fig. 6 and Appendix D.

## 8    Discussion

### 8.1 Verification of 3D wind recovery

Although not explicitly verified, the recovered 3D wind data were implicitly verified through the verification of recovered sonic temperature data because the recovery of sonic temperature data must rely on recovered 3D wind data [see Eqs. (33) and (40)] as the cross wind correction for sonic temperature needs 3D wind data (Liu et al., 2001). If 3D wind had not been well recovered, sonic temperature data could not have been recovered satisfactorily. The satisfactory recovery of sonic temperature data in this study implicitly verified the satisfactory recovery of 3D wind data.

### 8.2 Comparability of recovered to calculated sonic temperature

The recovered sonic temperature was sourced from the measurements of a fast response sonic anemometer and the calculated sonic temperature was sourced from the measurements of a slow response air temperature and relative humidity probe as well as IRGASON built-in barometer (see Appendix D); Therefore, the former reflected the fluctuations in sonic temperature at high frequency, the latter reflects such fluctuations not so high as the former. As such, a pair of recovered and
calculated sonic temperature values from simultaneous measurements (i.e., the same record in a time series data file) were not comparable. The difference between the pair is meaningless; therefore, the mean difference between recovered and calculated sonic temperature values over a half-hour period was used for their data comparison.

For better comparison, the difference was calculated from the recovered and calculated sonic temperature values temporally aligned in consideration of lag. The calculated sonic temperature from the air temperature, relative humidity, and
atmospheric pressure; which were measured using slow response sensors; was believed to be lagged behind recovered one in response to the fluctuations in sonic temperature. The lag time about 10 seconds was empirically found at the maximization of cross correlation of a time series of recovered sonic temperature to different time-lagged (i.e., time-shifted) series of calculated sonic temperature (Ibrom et al., 2007). The difference between recovered and calculated sonic temperature values was calculated using recovered sonic temperature without a time lag and calculated sonic temperature with a time lag of 10
seconds. The use of lag time might be unnecessary, but might make the comparison as reasonable as possible.

### 8.3 Recovered higher than calculated sonic temperature at lower temperature

See Fig. 7. Compared to calculated sonic temperature, the recovered sonic temperature from Eq. (40) is 0.81 ± 0.14 ℃ higher at -20.01 ℃ (Fig. 7a) and 0.38 ± 0.17 ℃ higher at -9.06 ℃ (Fig. 7b), however, at -1.90 ℃, even 0.45 ± 0.24 ℃ lower (Fig. 7c). This trend of difference with temperature may be related to the performance of sonic anemometer at
different temperature and the lower accuracy of temperature and humidity probe in a lower temperature range (Campbell Scientific Inc., 1990).

The sonic path lengths and geometry of sonic anemometer were measured at the manufacture environment of air temperature around 20 ℃ (i.e., manufacture temperature) and embedded into its firmware for field applications. However, above or below the manufacture temperature, the sonic path lengths must become, due to thermo-expansion or -contraction of sonic
anemometer structure, longer or shorter than those at manufacture temperature while the length values of sonic paths are unchanged inside firmware. As a result, the sonic anemometer could under- or over-estimate the speed of sound, thus sonic temperature. The under- or over-estimation may be insignificant when temperature is not much above or below the





manufacture temperature while the anemometer must work best around the manufacturer temperature. In the case of this study, the working air temperature for the sonic anemometer was as low as negative to -20 ℃, within which the sonic paths become shorter at some degree so that its measurement performance possibly was impacted. Although an assessment on the measurement performance of sonic anemometer at low or high air temperature could not be found in literature,

overestimation of the speed of sound from a sonic anemometer at centigrade of tens below manufacture temperature and thus sonic temperature is anticipated as shown in Fig. 7a to Fig. 7c.

Although, at different air temperature, the performance of the temperature and relative humidity probe and IRGASON built-in barometer, whose measurements are used to calculate the sonic temperature (see Appendix D), is more stable than a sonic anemometer although their accuracies are the best at 20 ℃, too, and become lower with temperature away from 20 ℃

(Campbell Scientific Inc., 1990). For example, HMP155A has an accuracy in air temperature to be ±0.1 ℃ at 20 ℃ and to be ±0.25 ℃ at -20 ℃ as well as an accuracy in relative humidity ($RH$) to be ±(1.0+0.008$RH$) in % at 20 ℃ and to be ±(1.2+0.012$RH$) in % at -20 ℃. The greater disagreement between recovered and calculated sonic temperature values at lower temperature in Fig. 7a may also be contributed by the fact that the lower the air temperature, the lower the accuracies of HMP155A and IRGASON built-in barometer.

**8.4 Radiation on calculated sonic temperature**

See Fig. 7c. Compared to the recovered sonic temperature using Eq. (40), the calculated sonic temperature was 0.45 ± 0.24 ℃ higher over a whole period of 12:00 to 12:30 and even 0.65 ±0.19 ℃ higher over a partial period of 12:15 to 12:27, which may be contributed in part by higher incoming solar radiation of 750 W m$^{-2}$ in short-wave on the radiation shield of HMP155A. As addressed in Appendix D, the calculated sonic temperature was sourced from the measurements of air

temperature and relative humidity from HMP155A as well as atmospheric pressure from IRGASON built-in barometer. The HMP155A housed inside a radiation shield (see Fig. 2) was subject to contamination from solar radiation. Even a radiation shield was used to shade HMP155A from sunlight, when such a shield was used, any heat generated from the shield under sunlight and the sensor under electronic power was dissipated inefficiently (Lin et al., 2001). As a result, the air and HMP155A sensing elements inside the shield were warmer than ambient air of interest. How warm the air is inside the

radiation shield depended on shield structure, ambient wind speed and other environmental conditions (Blonquist et al., 2009). In case of Fig. 7c at 750 W m$^{-2}$ of incoming short-wave radiation, a degree higher of air inside the radiation shield was not unusual (Lin et al., 2001). In our study, this higher air temperature could directly cause the overestimation of calculated sonic temperature (see Eq. D1 in Appendix D)

**8.5 Applicability of equations and algorithms in this study**

Any sonic anemometer is built as slender (e.g., < 1.00 cm in each diameter of CSAT3 six claws to hold individual sonic transducers) and light as possible to minimize its aerodynamic resistance to air flows and to maximize its stability on supporting infrastructure (e.g., tripod) to wind momentum load, which sacrifices its durability in keeping its geometrical shape; therefore, a sonic anemometer is easily deformed if not well cared for in transportation (e.g., the case in this study), installation, or other handlings. As shown in this study, a slight geometrical deformation, even changes of millimeters or less

of sonic path length (see Table A1 in Appendix A) could cause significant errors in 3D wind and, especially, in sonic temperature. According to our recalibration experience with 3D sonic anemometers at Campbell Scientific Incorporation, these cases as addressed in this study have been not unusual, but the equations and algorithms to recover the data measured by a deformed 3D sonic anemometer were not available. Since requisition of these datasets are expensive, thus their recovery would be the cost-effective and time-saving option.





The equations and algorithms in this study were developed based on the measurement working physics and sonic path geometry of IRGASON sonic anemometer. The physics is the same as those for other models of Campbell Scientific 3D sonic anemometers such as CSAT3, CSAT3A, and CSAT3B that are popularly used in the world (Horst et al., 2015). The sonic path geometry of IRGASON sonic anemometer, however, is different from other models in the assigned azimuth angle

of $1^{st}$ sonic path in the 3D coordinate system. This angle was assigned as 90 $^\circ$ in IRGASON sonic anemometer, but as 0 $^\circ$ in other models (Campbell Scientific Inc., 1998; 2010b; 2015). Even so, given the sonic path lengths and transfer matrixes of sonic anemometer that were measured and determined in manufacture or calibration process [$d_i$ in Eq. (12) and $\mathbf{A}$ in Eq. (15)] and in recalibration process after the use in its geometrical deformation state [$d_{Ti}$ in Eqs. (13), (33), and (40) and $\mathbf{A_T}$ in Eqs. (14) and (16)], the equations and algorithms from this study are applicable to all models of Campbell Scientific 3D sonic

anemometers (see Fig. 6). The derivation procedures and even equations based on the measurement working physics are applicable as a reference to the development of the equations and algorithms to recover the data measured using other brands of 3D sonic anemometers that incurred deformations or to studies on similar topics.

## 9 Conclusion remarks

An IRGASON 3D sonic anemometer (SN: 1131) which was geometrically deformed by impact during transportation to

Antarctica from China early 2015. To fulfill the field measurement plans for the year, it had to be deployed there in the Zhongshan Station until early 2016 when it was replaced in the field with another IRGASON provided by the manufacturer and was returned to its manufacturer, Campbell Scientific Incorporation, for recalibration through the re-measurements of its sonic path geometry (i.e., lengths and angles), re-determination of transfer matrix, and update of operating system. To recover the 3D wind and sonic temperature data measured by this sonic anemometer in its deformed state before the

recalibration, equations and algorithms were developed and implemented into a software package: "Sonic Data Recovery for IRGASON/CSAT3/A/B Used in Geometrical Deformation after Production/Calibration" as shown in Fig. 6. Given two sets of sonic path lengths and two transfer matrixes of sonic anemometer that were measured and determined in manufacture/calibration process and also in recalibration process after the use in its deformed state, the data measured by the IRGASON 3D sonic anemometer even in its deformed state were recovered as if measured by the same anemometer

recalibrated immediately after its deformation.

Inside a Campbell Scientific sonic anemometer, the transducer shadow correction for 3D wind (Kaimal and Finnigan, 1994) is a programmable option to a user; however, the crosswind correction for sonic temperature (Liu et al., 2001) is internally applied as default by its firmware. In a case of transducer shadow correction in option, the 3D wind data are recovered using Eqs. (10) to (16). If not, Eqs. (15), (12), (13), and (16) are sequentially used. Based on the data from the recovery process of

3D wind, the sonic temperature data are recovered using Eqs. (9), (32), (38), and (40); therefore, the satisfactory recovery for both 3D wind data and sonic temperature can be reflected eventually by the satisfactory of sonic temperature data recovery.

The software based on the equations and algorithms from this study can recover the 3D wind data with/without the transducer shadow correction inside the sonic anemometer and sonic temperature data with crosswind correction also inside the sonic anemometer. It was verified by comparing the recovered to calculated sonic temperature data (see Appendix D). As

shown in Fig. 7, the recovered data of sonic temperature using Eq. (33) and Eq. (40) were compared to the calculated sonic temperature of three representative values over the range of measured sonic temperature from -20.01 to -1.90 ℃. The difference of 9.60 to 8.93 ℃ between unrecovered and calculated sonic temperature (i.e., unrecovered minus calculated) was narrowed by Eq. (40) to 0.81 to -0.45 ℃ (i.e., recovered minus calculated), which was satisfactory for measurements of sonic anemometer below 0 to -20 ℃. After verification, the software was used to recover the data measured by the IRGSON



(SN: 1131) 3D sonic anemometer in its deformed state from May 2015 to January 2016. The eight-month data were recovered using three days of one engineer's time. Further using EddyPro 6.2.0 (Li-Cor Inc., 2016), the recovered data were further processed for the fluxes of $CO_2/H_2O$, sensible heat, and momentum. The data quality (Foken et al., 2012) mostly range in 1 to 3 and the energy closure without considering surface heat flux into ice were >83% when friction velocity was >

0.2 m s$^{-1}$.

The use of a deformed 3D sonic anemometer is a practical case. If the data from such a use cannot be recovered, the requisition of these data are expensive and their recovery would be a cost-effective and time-saving option. The equations, algorithms, and software are applicable to all models of Campbell Scientific 3D sonic anemometers such as CSAT3, CSAT3A, and CSAT3B that are popularly used around the world. The derivation procedures and even equations based on

the measurement working physics of sonic anemometers are applicable as a reference to the development of the equations and algorithms to manage the data measured using other brands of 3D sonic anemometers or recover the data measured by an anemometer in its deformed state.

**Appendix A Transform matrixes**

In micrometeorological applications, the wind speeds are expressed in a three-dimensional (3D) orthogonal coordinate system of instrument or natural wind, but a sonic anemometer measures flow velocities along its three non-orthogonal sonic paths (i.e. situated non-orthogonally each other, see Figs. 1 and A1); therefore, for applications, the flow velocities along the three sonic paths need to be transformed into a 3D right-handed orthogonal coordinate system in reference to the geometry of sonic anemometer as shown in Fig. A1 (i.e., the 3D orthogonal instrument coordinate system). Given $u_x$ and $u_y$ are two

horizontal velocities in the $x$- and $y$-direction, respectively, and $u_z$ is vertical velocity in the $z$-direction (Fig. A1); $x$, $y$, and $z$ are the three coordinate axes in the 3D orthogonal instrument coordinate system. This system is defined with the $x$-$y$ plain parallel to the anemometer bulb-leveled instrument plain, with the 1$^{st}$ sonic path on the $y$-$z$ plain, and with origin in the center of measurement volume. A flow speed along the $i$th ($i$ = 1, 2, or 3) sonic path is a combination of component velocities along the path from $u_x$, $u_y$, and $u_z$; given by:

$$u_i = \left(u_x \cos\varphi_i + u_y \sin\varphi_i\right)\sin\theta_i + u_z \cos\theta_i \tag{A1}$$

where $\theta_i$ and $\varphi_i$ are the zenith and azimuth angles of the $i$th sonic path in the 3D orthogonal instrument coordinate system. In this system (see Fig. A1), given the 1$^{st}$ sonic path has an azimuth angle of $\varphi_1$ equal to 90 ° as fixed on the $x$-$y$ plain, Eq. (A1) can be expressed in a matrix form of

$$
\begin{bmatrix} u_1 \\ u_2 \\ u_3 \end{bmatrix} =
\begin{bmatrix}
0 & \sin\theta_1 & \cos\theta_1 \\
\sin\theta_2\cos\varphi_2 & \sin\theta_2\sin\varphi_2 & \cos\theta_2 \\
\sin\theta_3\cos\varphi_3 & \sin\theta_3\sin\varphi_3 & \cos\theta_3
\end{bmatrix}
\begin{bmatrix} u_x \\ u_y \\ u_z \end{bmatrix} =
\mathbf{A^{-1}}
\begin{bmatrix} u_x \\ u_y \\ u_z \end{bmatrix} \tag{A2}
$$

30   where $\mathbf{A}$ is a matrix expressing the flow speeds along the three non-orthogonal sonic paths in the 3D orthogonal instrument coordinate system. Nominally for the sonic paths of IRGASON, $\theta_1$, $\theta_2$, and $\theta_3$ are all 30 ° and $\varphi_2$ and $\varphi_3$ are 330 ° and 210 ° (see Fig. A1). Given $\varphi_1$ = 90 °, these angles are calculated using measured data from Coordinate Measurement Machine and, along with the sonic path lengths, are listed in Table A1 for IRGASON Serial Number of 1131 before and after its geometrical deformation.





Table A1:The lengths, zenith angles, and azimuth angles of sonic paths in IRGASON (Serial Number: 1131)
instrument coordinate system before and after its geometrical deformation (measured using Coordinate
Measurement Machine in September 09, 2014 before the deformation and in March 06, 2016 after use in deformation)

|  | Geometrical deformation | 1st path $i = 1$ | 2nd path $i = 2$ | 3rd path $i = 3$ |
|---|---|---|---|---|
| Path length | before | 11.6486 | 11.5240 | 11.4968 |
| ($d_i$ in cm) | after | 11.6160 | 11.1245 | 11.3548 |
| Zenith angle | before | 29.935379 | 29.026608 | 29.612041 |
| ($\theta_i$ in °) | After | 29.925878 | 25.226585 | 28.772601 |
| Azimuth angle | Before | 90.000000 | 329.527953 | 206.80477 |
| ($\varphi_i$ in °) | After | 90.000000 | 324.736084 | 209.23382 |

Using the data in this table, matrix $\mathbf{A}$ in Eq. (A2) and its inversion $\mathbf{A}^{-1}$ for this IRGASON before its geometric deformation
(i.e., as used in IRGASON firmware although not valid in the field after deformation) are given

$$\mathbf{A} = \begin{bmatrix} 0.034785 & 1.142665 & -1.183914 \\ 1.365505 & -0.696580 & -0.660515 \\ 0.367627 & 0.401124 & 0.380356 \end{bmatrix} \qquad (A3)$$

and

$$\mathbf{A}^{-1} = \begin{bmatrix} 0.00000 & 0.499023 & 0.866589 \\ 0.418196 & -0.246062 & 0.874394 \\ -0.441030 & -0.222826 & 0.869391 \end{bmatrix} \qquad (A4)$$

After the IRGASON geometrical deformation, matrix $\mathbf{A}$ became:

$$\mathbf{A_T} = \begin{bmatrix} 0.006035 & 1.276412 & -1.323287 \\ 1.363991 & -0.724862 & -0.600545 \\ 0.368690 & 0.417250 & 0.345690 \end{bmatrix} \qquad (A5)$$

where subscript $T$ indicates "True" because, after IRGASON deformation, it should be used in the field although it was not
used. The inversion of this matrix is given as

$$\mathbf{A_T}^{-1} = \begin{bmatrix} 0.000000 & 0.498879 & 0.866672 \\ 0.347992 & -0.246063 & 0.904629 \\ -0.420029 & -0.235072 & 0.876537 \end{bmatrix} \qquad (A6)$$

Matrixes $\mathbf{A}^{-1}$, $\mathbf{A}_T$, and $\mathbf{A}_T^{-1}$ were used for our data recovery and $\mathbf{A}$ was also used in the firmware inside the IRGASON sonic
anemometer.

**Appendix B Iteration algorithm for sonic transducer shadow corrections**

Given transform matrix $\mathbf{A}$, using Eq. (5), the measured wind vector $\begin{bmatrix} u_1 & u_2 & u_3 \end{bmatrix}'$ along the sonic paths is transformed to

the wind vector in the 3-dimensioanl orthogonal instrument coordinate system $\begin{bmatrix} u_x & u_y & u_z \end{bmatrix}'$. Subsequently, $U_T$ is

calculated using Eq. (9). Replace $u_{Ti}$ with $u_i$ under the square root in the right hand of Eq. (8), an approximate equation for
the 1st iteration is given:





$$u_{Ti\_1} \approx \frac{u_i}{0.84 + 0.16\dfrac{\sqrt{U_T^2 - u_i^2}}{U_T}} \qquad \text{(B1)}$$

where subscript $i$ is 1, 2 or 3 and subscript $\_1$ of $u_{Ti}$ indicates that it is calculated from the 1$^{st}$ iteration.

**1$^{st}$ iteration**

Equation (B1) is used for sonic transducer shadow corrections in the first iteration.

**2$^{nd}$ iteration**

$$\begin{bmatrix} u_x \\ u_y \\ u_z \end{bmatrix} = \mathbf{A} \begin{bmatrix} u_{T1\_1} \\ u_{T2\_1} \\ u_{T3\_1} \end{bmatrix} \qquad \text{(B2)}$$

Using Eq. (9), $U_T$ is recalculated. Replace $u_i$ with $u_{Ti\_1}$ under the square root in the right hand of Eq. (B1), an approximate equation for the 2$^{nd}$ iteration is given:

$$u_{Ti\_2} = \frac{u_i}{0.84 + 0.16\dfrac{\sqrt{U_T^2 - u_{Ti\_1}^2}}{U_T}} \qquad \text{(B3)}$$

**3$^{rd}$ iteration**

………………

**n$^{th}$ iteration**

$$\begin{bmatrix} u_x \\ u_y \\ u_z \end{bmatrix} = \mathbf{A} \begin{bmatrix} u_{T1\_m} \\ u_{T2\_m} \\ u_{T3\_m} \end{bmatrix} \qquad \text{(B4)}$$

where subscript $m = n - 1$. Using Eq. (9), $U_T$ is also recalculated. Similar to the calculation for $u_{Ti\_2}$, $u_{Ti\_n}$ is calculated using

equation

$$u_{Ti\_n} = \frac{u_i}{0.84 + 0.16\dfrac{\sqrt{U_T^2 - u_{Ti\_m}^2}}{U_T}} \qquad \text{(B5)}$$

to ensure that the difference in $u_x$, $u_y$, or $u_z$ between last and previous iterations are $< 1\,\mathrm{mm}\ \mathrm{s}^{-1} \approx 1.96\sigma$ where $\sigma$ is the maximum precision (i.e. standard deviation at constant wind) among $u_x$, $u_y$, and $u_z$ (Campbell Scientific Inc., 2010b). Our numerical testes within the measurement ranges in $u_x$, $u_y$, and $u_z$ concluded that the iterations mostly converged at $n = 2$ and

all at $n \leq 3$.





**Appendix C: MATLAB code: Sonic data recovery for IRGASON/CSAT3/A/B used in geometrical deformation after production/calibration (Code lines were formatted for readability and the dialog interface related lines was removed for proprietary):**

% sonicdatarecovery Sonic Data Recovery for IRGASON/CSAT3/A/B Used in Geometrical Deformation after

Production/Calibration

**%Syntax:**

% [*ucm, Tcs, Tcs_ad*] = sonicdatarecovery(*um, Ts,* A_inversion, AT_inversion, *di, dTi, shadow_correction_flag*)

**% Inputs:**

% *um*        Measured 3D wind speeds in the orthogonal instrument coordinate system (OCS)

% *Ts*         Measured sonic temperature

% A        Matrix of sonic to OCS before geometrical deformation

% AT       Matrix of soninc to OCS after geometrical deformation

% *di*        Sonic path length before geometrical deformation (i =1,2, or 3)

% *dTi*      Sonic path length after geometrical deformation (i =1,2, or 3)

**% Constants**

shadow_correction_flag =1; %%Shadow correction has been done (=1) or not (=0) inside firmware

gama_d=1.4003;      %% the ratio of dry air specific heat at constant pressure to that at constant volume

Rd=287.04;        %% gas constant for dry air

RV=4.61495e-4;     %% gas constant for water

Av=60.064621; Bv=60.973392; Cv=60.387959; Ah=0.000000; Bh=59.527953; Ch=63.195226;

Avt=60.074122; Bvt=64.773415; Cvt=61.227399; Aht=0.000000; Bht=54.736084; Cht=60.766176;

**% Browse to the raw data file directory to load the files in a batch**

RAW=dlmread('C:\xxxx\TOA5_7720.ts_data_2015_05_09_1639_raw.dat',',', 4, 1);

**% Extract sonic anemometer and other meteorological data**

UX=RAW(:,2); UY=RAW(:,3); UZ=RAW(:,4);

TRAW=RAW(:,5); H2O=RAW(:,8); Temp=RAW(:,10); P=RAW(:,11);

amb_e=RV.*H2O.*(Temp+273.15); TS_emp=(Temp+273.15).*(1+0.32*amb_e./P)-273.15;

**% Load transform matrix of eq. (A2) and data of Table A1 before geometrical deformation**

The1=((90-Av)/180)*pi; The2=((90-Bv)/180)*pi; The3=((90-Cv)/180)*pi;

Phi1=((90-Ah)/180)*pi; Phi2=((270+Bh)/180)*pi; Phi3=((270-Ch)/180)*pi;

A_inversion=[0  sin(The1) cos(The1); sin(The2)*cos(Phi2) sin(The2)*sin(Phi2) cos(The2);

sin(The3)*cos(Phi3) sin(The3)*sin(Phi3) cos(The3)];

A=A_inversion^(-1); d=[11.6486;11.5240;11.4968];

**% Load transform matrix of eq. (A4) and data of Table A1 after geometrical deformation**

The1=((90-Avt)/180)*pi; The2=((90-Bvt)/180)*pi; The3=((90-Cvt)/180)*pi;

Phi1=((90-Aht)/180)*pi; Phi2=((270+Bht)/180)*pi; Phi3=((270-Cht)/180)*pi;

AT_inversion=[0  sin(The1) cos(The1); sin(The2)*cos(Phi2) sin(The2)*sin(Phi2) cos(The2);sin(The3)*cos(Phi3)

sin(The3)*sin(Phi3) cos(The3)];

AT= AT_inversion ^(-1); dT=[11.6159;11.1245;11.3548];

**%Procedure to recover 3D wind data: Get measured flow speeds along each of 3 sonic paths**

[mRaw,nRaw]=size(RAW);

for i=1:mRaw;





um=[UX(i);UY(i);UZ(i)];

**%With transducer shadow corrections (TSC):**

UT=(um(1)^2+um(2)^2+um(3)^2)^(1/2);  %%Calculate the total wind magnitude

if isequal(shadow_correction_flag, 1)    %%TSC has been done (=1) inside firmware

u=A_inversion*um;    %%% Calculate the vector of the three flow speeds using Eg (10)

ut1(1)=u (1)/(0.84+0.16.*((UT^2-u (1)^2)^(1/2))./UT); %%% Eq (11) to recover flow speed along sonic path 1 before TSC

ut2(1)=u (2)/(0.84+0.16.*((UT^2-u (2)^2)^(1/2))./UT); %%% Eq (11) to recover flow speed along sonic path 2 before TSC

ut3(1)=u (3)/(0.84+0.16.*((UT^2-u (3)^2)^(1/2))./UT); %%% Eq (11) to recover flow speed along sonic path 3 before TSC

uc=[ut1.*(dT (1)./d(1));ut2.*(dT(2)./d(2));ut3.*(dT(3)./d(3))]; %%Eq (13)

uts1=ut1; uts2=ut2; uts3=ut3;

%%Corrected 3D wind speed

um_c=AT*uc; %%Eq (16)

**%Iteration algorithm of sonic TSC (Appendix B) for corrected data**

UT_C=(um_c (1)^2+um_c (2)^2+um_c (3)^2)^(1/2); %%% Total wind magnitude

%% 1st iteration

uct1=uc (1)/(0.84+0.16.*((UT^2-uc (1)^2)^(1/2))./UT); %%%flow speed 1

uct2=uc (2)/(0.84+0.16.*((UT^2-uc (2)^2)^(1/2))./UT); %%%flow speed 2

uct3=uc (3)/(0.84+0.16.*((UT^2-uc (3)^2)^(1/2))./UT); %%%flow speed 3

%% 2nd iteration

for q=2:5; %%% 5 steps of iterations after the 1st iteration are adequate

%%%TSC for flow speed 3

uct_m=[uct1(q-1);uct2(q-1);uct3(q-1)];        %%%% Vector of three path flow speeds

um_C=AT*uct_m;            %%%%Vector in 3D orthogonal system

UT_C=(um_C (1)^2+um_C (2)^2+um_C (3)^2)^(1/2); %%%%Total wind magnitude, again

uct3(q)=uc (3)/(0.84+0.16.*((UT_C^2-uct3 (q-1)^2)^(1/2))./UT_C); %%% TSC for flow speed 3

%%% TSC for flow speed 2

uct_mm=[uct1(q-1);uct2(q-1);uct3(q)];       %%%%Vector of three flow speeds, again

um_C=AT*uct_mm;          %%%% Vector in 3D orthogonal system, again

UT_C=(um_C (1)^2+um_C (2)^2+um_C (3)^2)^(1/2); %%%% Recalculated the total wind magnitude

uct2(q)=uc (2)/(0.84+0.16.*((UT_C^2-uct2 (q-1)^2)^(1/2))./UT_C); %%% TSC for flow speed 2

%%%TSC for flow speed 1

uct_mm=[uct1(q-1);uct2(q);uct3(q)];        %%%%Vector of three flow speeds, again

um_C=AT*uct_mm;         %%%% Vector in 3D orthogonal system

UT_C=(um_C (1)^2+um_C (2)^2+um_C (3)^2)^(1/2); %%%% Total wind magnitude, again

uct1(q)=u (1)/(0.84+0.16.*((UT_C^2-uct1 (q-1)^2)^(1/2))./UT_C); %%%TSC for flow speed 1

%% Judge the steps of iterations

uct_n=[uct1(q);uct2(q);uct3(q)];       %%%Vector from current iteration

ABS_C=uct_n-uct_m;        %%%%Difference between two iterations

   %%%%Exit condition

if(abs(ABS_C(1))<=0.001&&abs(ABS_C(2))<=0.001&&abs(ABS_C(3))<=0.001);

**%Finalize recovered 3D wind speed**





```
     ucm=AT*uct_n;                    %% Eq (14)
       ucts1=uct1(q); ucts2=uct2(q); ucts3=uct3(q);
     break;                    %% %Exit iterations
       end
end
     else
```

**%Recover 3D wind data without TSC**

```
     u=A_inversion*um;          %%% Inverse to have the three path flow speeds using Eq (10)
     uc=[dT(1)./d(1).*u(1); dT(2)./d(2).*u(2); dT(3)./d(3).*u(3)]; %%%Correction
ucm=AT*uc;                       %%%3D orthogonal data after correction
     uts1=uc(1); uts2=uc(2); uts3=uc(3);
     ucts1=ucm(1); ucts2=ucm(2); ucts3=ucm(3);
     end
```

**%Procedure to recover sonic temperature data**

```
Ts=TRAW(i);
     UcT= (ucm (1)^2 + ucm (2)^2 + ucm (3)^2)^(1/2);      %% Total wind
     CT2 = gama_d*Rd*(Ts + 273.15);             %% Eq (32)
     DELTUcT2 = UcT^2 - UT^2;
     DELTucT21 = ucts1^2 - uts1^2;  DELTucT22=ucts2^2 - uts2^2; DELTucT23=ucts3^2 - uts3^2;
DELTC21=(CT2 - UT^2 + uts1^2)*((dT(1)^2 - d(1)^2)/d(1)^2); %% Eq (30)
     DELTC22=(CT2 - UT^2 + uts2^2)*((dT(2)^2 - d(2)^2)/d(2)^2); %% Eq (30)
     DELTC23=(CT2 - UT^2 + uts3^2)*((dT(3)^2 - d(3)^2)/d(3)^2); %% Eq (30)
     AAA=(DELTC21 + DELTUcT2 - DELTucT21);
     BBB=(DELTC22 + DELTUcT2 - DELTucT22);
CCC=(DELTC23 + DELTUcT2 - DELTucT23);
     DDD=(AAA + BBB + CCC);
     EEE=3*gama_d*Rd;
     Tcs=Ts+(DDD/EEE);                %% Eq (33)
     DELTCT21_ad=CT2*2*(1-dT(1)/d(1));           %% Eq (38)
DELTCT22_ad=CT2*2*(1-dT(2)/d(2));           %% Eq (38)
     DELTCT23_ad=CT2*2*(1-dT(3)/d(3));           %% Eq (38)
     AAA_ad=((dT(1)^2-d(1)^2)/d(1)^2)*(CT2-(DELTCT21_ad+((DELTCT21_ad+DELTCT22_ad+DELTCT23_ad)/3))-
            UT^2+uts1^2)+DELTUcT2-DELTucT21;
     BBB_ad=((dT(2)^2-d(2)^2)/d(2)^2)*(CT2-(DELTCT22_ad+((DELTCT21_ad+DELTCT22_ad+DELTCT23_ad)/3))-
35          UT^2+uts2^2)+DELTUcT2-DELTucT22;
     CCC_ad=((dT(3)^2-d(3)^2)/d(3)^2)*(CT2-(DELTCT23_ad+((DELTCT21_ad+DELTCT22_ad+DELTCT23_ad)/3))-
            UT^2+uts3^2)+DELTUcT2-DELTucT23;
     DDD_ad=(AAA_ad + BBB_ad + CCC_ad);
     Tcs_ad=Ts+(DDD_ad/EEE); %% Eq (40)
Data_recovery(i,1)= ucm(1);     %%Recovered 3D wind speed in x-direction
     Data_recovery(i,2)= ucm(2);     %% Recovered 3D wind speed in y-direction
```



Data_recovery(i,3)= ucm(3);    %% Recovered 3D wind speed in z-direction

Data_recovery(i,4)= Tcs;    %% Recovered sonic temperature from raw Ts using Eq (33)

Data_recovery (i,5)= Tcs_ad;    %% Recovered sonic temperature from raw Ts using Eq (40)

Data_recovery (i,6)= TS_emp(i); %% Recovered air temperature using Eq (D1)

5  Data_recovery (i,7)= TRAW(i);   %% Raw data of Ts

end

**Appendix D Sonic temperature from air temperature, relative humidity, and atmospheric pressure**

In case that air temperature ($T$ in °C), relative humidity ($RH$ in %), and atmospheric pressure ($P$ in kPa) are measured in the field, sonic temperature ($T_s$ in °C) can be calculated using the well-known equation (Kaimal and Gaynor, 1991):

$$T_s = (T + 273.15)(1 + 0.32\frac{e}{P}) - 273.15 \tag{D1}$$

where $e$ is air water vapor pressure (kPa) and can be computed from $T$, $RH$, and $P$ as following.

Given $T$ and $P$, saturated water vapor pressure ($e_s$ in kPa) can be calculated using Buck (1981):

$$e_s = \begin{cases} 0.61121\exp(\dfrac{17.368T}{T+238.88})f_w(T,P) & T \geq 0 \\[3mm] 0.61121\exp(\dfrac{17.966T}{T+247.15})f_w(T,P) & T < 0 \end{cases} \tag{D2}$$

where $f_w(T, P)$ is the enhancement factor:

$$f_w(T,P) = 1.00041 + P\left[3.48 \times 10^{-5} + 7.4 \times 10^{-9}\left(T + 30.6 - 0.38P\right)^2\right] \tag{D3}$$

Using the definition of air relative humidity, air water vapor pressure is given by:

$$e = e_s \frac{RH}{100} \tag{D4}$$

Submit the measured $T$ and $P$ as well as the calculated $e$ into Eq. (D1), the sonic temperature can be calculated.

**Acknowledgment**

20  This study was supported by the National Natural Science Foundation of China (Grant #: 41376005, 41406218, 41505004, and 31200432). We thank the Chinese Arctic and Antarctic Administration and the Polar Research Institute of China for their field logistical support; Campbell Scientific Incorporation and Beijing Techno Solutions Limited for their customer supports; Steve Harston and Antoine Rousseau for technical graphic work; Carolyn Ivans, Bo Zhou, Mark Blonquist, and Hayden Mahan for their English polishing.

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





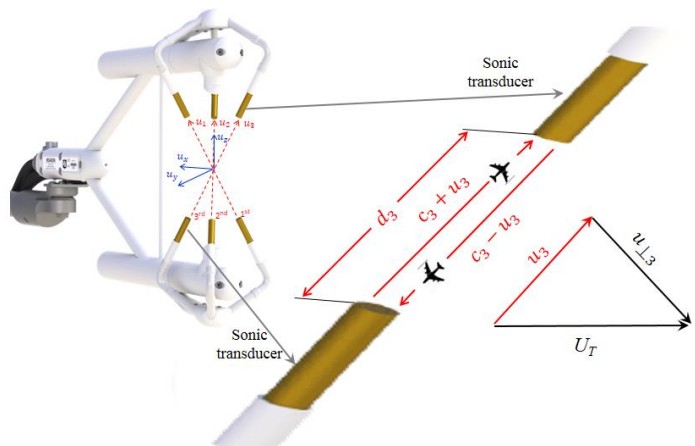

**Figure 1 Diagram of IRGASON for the three sonic measurement paths (red dash lines) along which ultrasonic signals fly and the three dimensional (3D) right-handed orthogonal instrument coordinate system (blue lines) in which 3D wind is expressed (i.e. $u_1$, $u_2$, and $u_3$ are the flow speeds along the 1st, 2nd, and 3rd sonic paths, respectively. These three flow speeds are expressed as $u_x$, $u_y$, and $u_z$ in this 3D instrument coordinate system; $d_3$ is the 3rd sonic path length; $c_3$ is the measured speed of sound along the 3rd sonic path; and $U_T$ is the total flow vector whose magnitude is equal to $\sqrt{u_3^2 + u_{\perp 3}^2}$ or $\sqrt{u_x^2 + u_y^2 + u_z^2}$ ).**




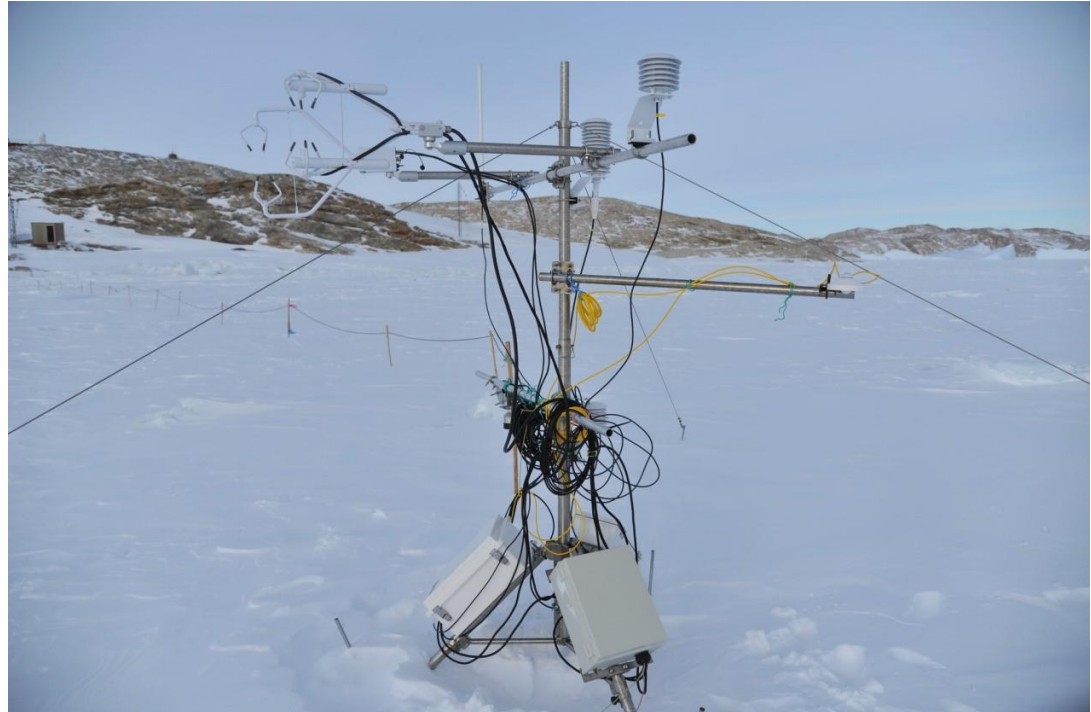

**Figure 2: The eddy-covariance station located in the coastal landfast sea ice area of Antarctica Zhongshan Station (69 °22′ S, 76 °22′ E). It was configured with IRGASON integrated CO$_2$/H$_2$O open-path gas analyzer and three-dimensional sonic anemometer, CNR4 4-Way Net Radiometer, HMP155A air temperature and relative humidity probe, and SI-111 infrared radiometer.**

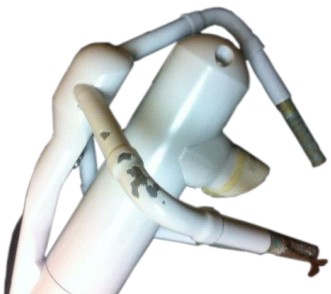

**Figure 3: Painting off as apparently impacted on the knuckle of side claw (i.e., 1[st] sonic path) among the top three sonic transducer claws of IRGASON sonic anemometer.**





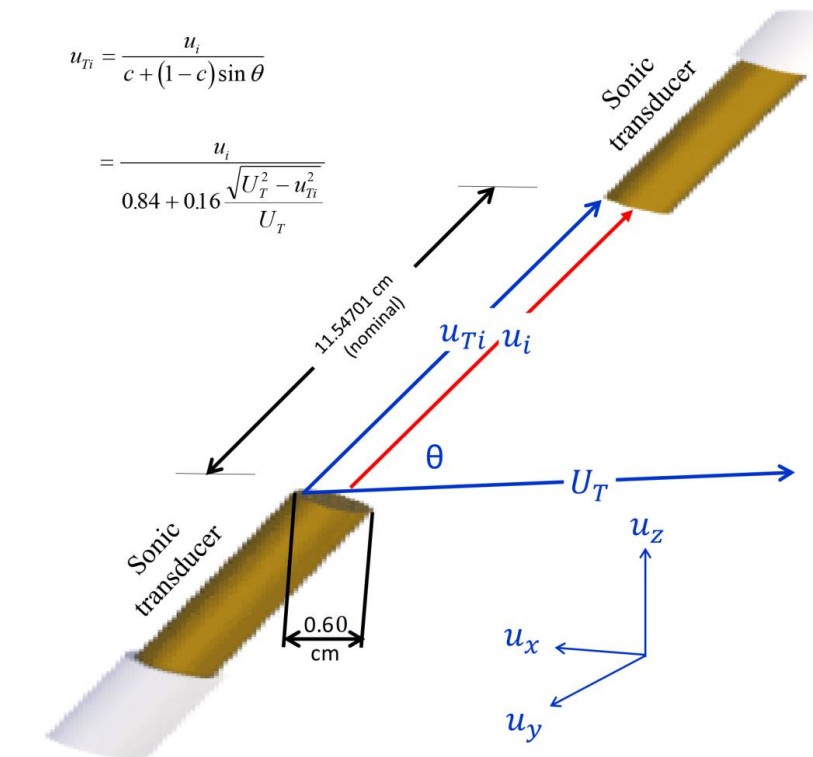

**Figure 4: Sonic transducer shadowing** [Along the *i*th (*i* = 1, 2, or 3) sonic path between the two sonic transducers, $u_i$ is the measured magnitude of flow vector whose true magnitude is $u_{Ti}$; $u_{\perp i}$ is the flow speed normal to the $i^{\text{th}}$ sonic path; $u_x$, $u_y$, and $u_z$ are the wind speeds expressed in the three-dimensional orthogonal instrument coordinate system; and $\alpha_i$ is the angle between sonic path *i* and the total flow vector ($U_T$) equal to $\sqrt{u_i^2 + u_{\perp i}^2}$ or $\sqrt{u_x^2 + u_y^2 + u_z^2}$ ]. See Wyngaard and Zhang (1985) and Kaimal and Finnigan (1994) for the equation to calculate $u_{Ti}$.




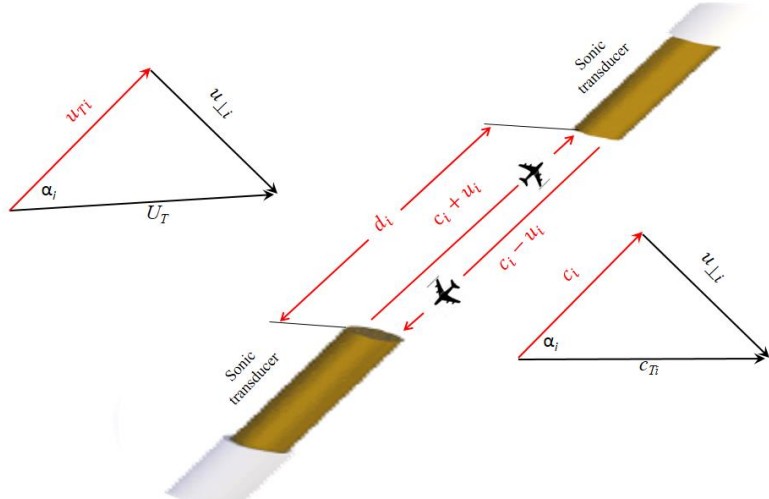

**Figure 5: Crosswind on speed of sound.** Along the $i$th ($i$ = 1, 2, or 3) sonic path between the two sonic transducers, $u_i$ is the measured magnitude of flow vector whose true magnitude is $u_{Ti}$, and $c_i$ is measured speed of sound; $u_{\perp i}$ is the crosswind vector

5    normal to sonic path $\underline{i}$; $U_T$ is the magnitude of total flow vector whose magnitude is equal to $\sqrt{u_i^2 + u_{\perp i}^2}$ or $\sqrt{u_x^2 + u_y^2 + u_z^2}$

where $u_x$, $u_y$, and $u_z$ are the wind speeds in the three-dimensional right-handed orthogonal instrument coordinate systems; $c_{Ti}$ is the true speed of sound; and $\alpha_i$ is the angle between sonic path $i$ and the total flow vector.



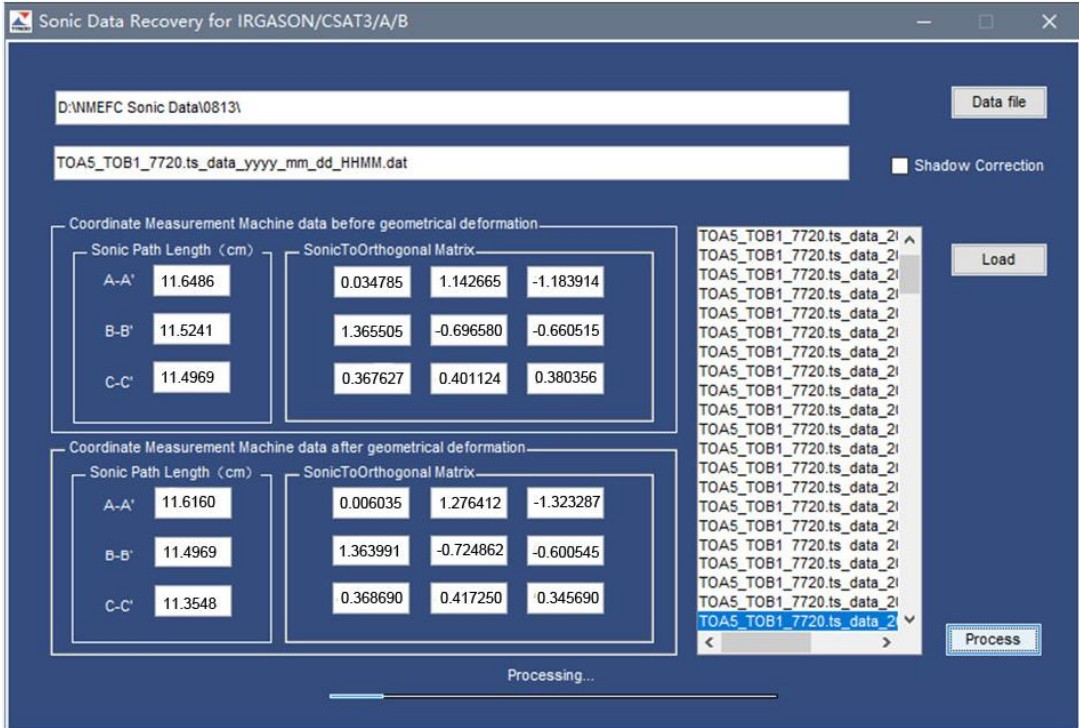

5    **Figure 6: Dialogue interface of software: Sonic Data Recovery for IRGASON/CSAT3/A/B Used in Geometrical Deformation after Production/Calibration.**



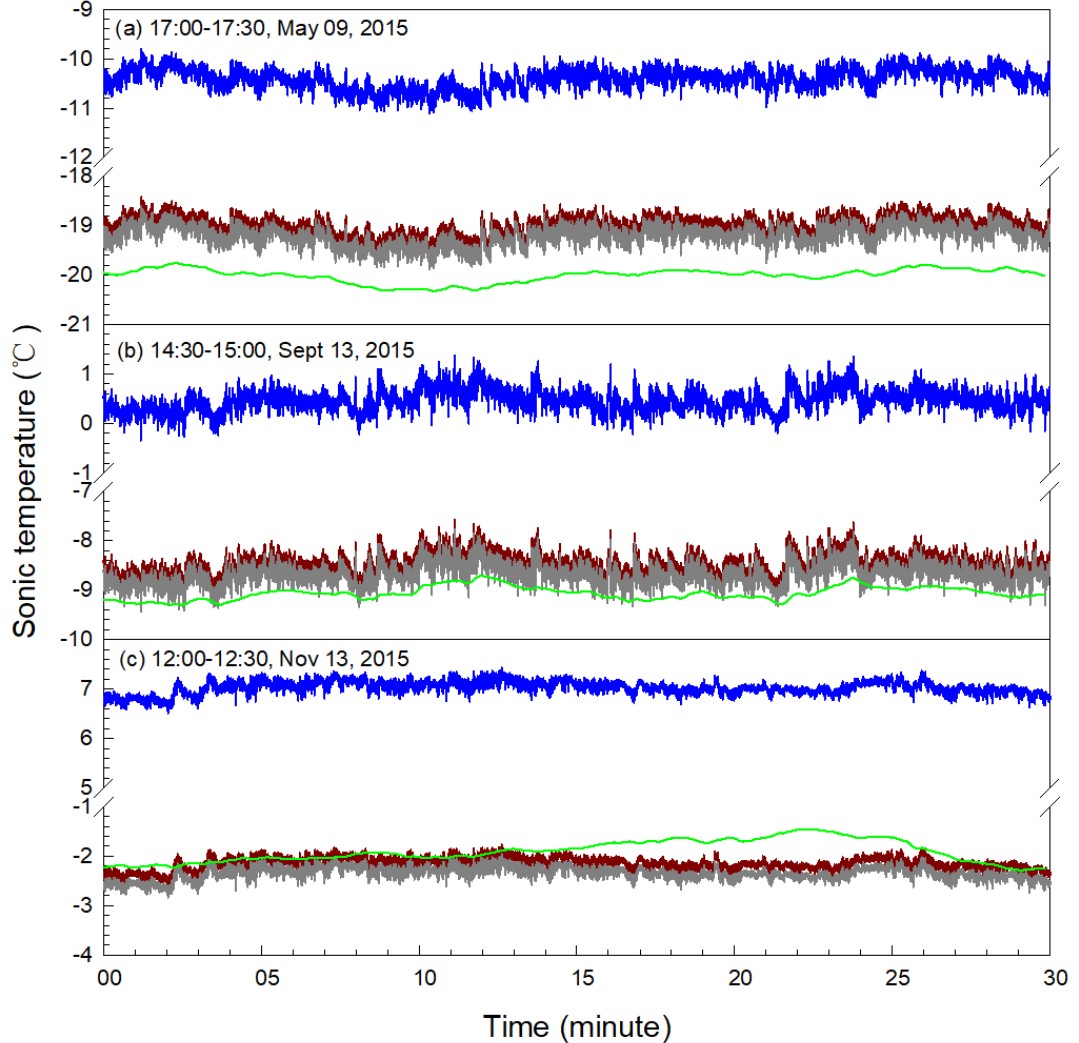

**Figure 7: Verification of sonic temperature ($T_s$) recovered against calculated (see Appendix D) from the air temperature ($T$), relative humidity ($RH$), and atmospheric pressure ($P$) that were measured using a HMP155A air temperature and relative humidity probe as well IRGASON built-in barometer {          $T_s$ measured by the IRGASON sonic anemometer in geometrical deformation (raw $T_s$),          $T_s$ recovered from raw $T_s$ using equation (33),          $T_s$ recovered also from raw $T_s$ using equation (40) [i.e., adjusted equation (33)],          $T_s$ calculated from $T$, $RH$ and $P$}.**



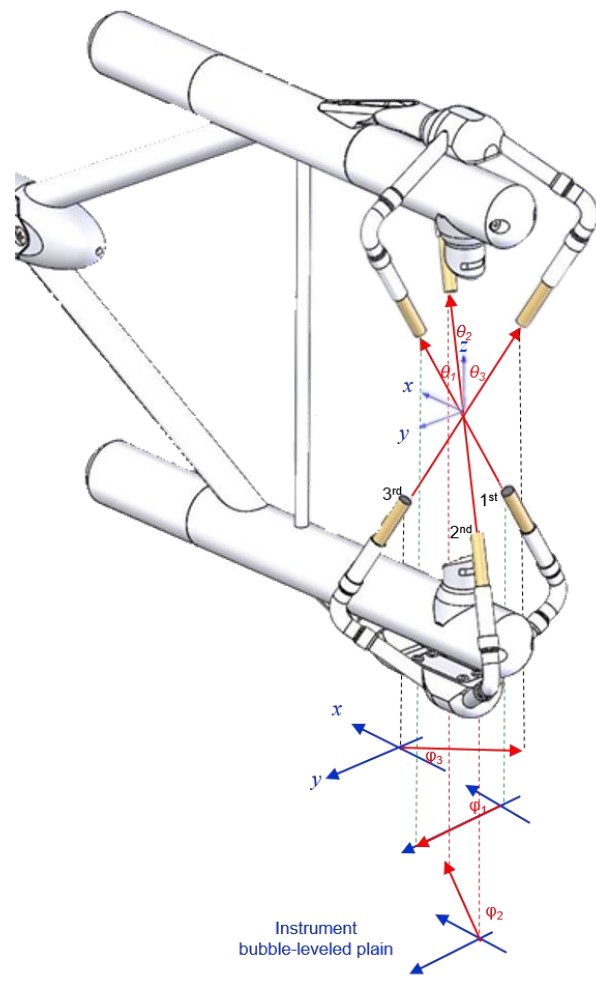

**Figure A1: IRGASON sonic path angle geometry in the three-dimensional right-handed instrument coordinate system of *x*, *y*, and *z* (Blue arrows are coordinates; a red arrow between a pair of sonic transducers is the sonic path vector whose direction is defined for air flow direction, a red arrow below the IRGASON is the projection of the corresponding sonic path vector on the *x-y* plain,**
5   **i.e. instrument bubble-leveled plain. As indicated by their subscript of 1, 2, or 3 for the 1st, 2nd, or 3rd sonic path, $\theta_1$, $\theta_2$, and $\theta_3$ are their zenith angles and $\varphi_1$, $\varphi_2$, and $\varphi_3$ are their azimuth angles)**