# Peer review of "Recovery of the 3-dimensional wind and sonic temperature data from a physically deformed sonic anemometer"

_Atmospheric Measurement Techniques, 2018_

## Referee Comment (RC1) · Anonymous Referee #1 · 28 Jul 2018

Title: Recovery of the 3-dimensional wind and sonic temperature data from a sonic anemometer physically deformed away from manufacture geometrical settings

Authors: Xinhua Zhou1,2,3, Qinghua Yang1, Xiaojie Zhen4, et al. Atmos. Meas. Tech. Discuss, doi: 10.5194/amt-2018-92

Summary:

=======

This paper describes the recovery of data from a sonic anemometer which was physi-

cally deformed during the shipping/installation process. The subject matter is appropriate for the AMT journal and the topic is practical and interesting. It is a useful to see the equations laid out and a practical application of "reverse engineering". With that said, I have one major complaint–the mean wind and temperature are good indications of the quality of the sonic data. However, the strength of a sonic anemometer is to measure wind fluctuations and fluxes. The important quantities that should be examined are the effect of the deformation on the fluctuations and fluxes (this is only vaguely described in terms of the surface energy balance on p. 16, but no mention of how the fluxes from the deformed instrument were affected). This important aspect of the study should be included before publication. Also, though the English has been improved, there are still many spots where further improvement is needed (I list a few below).

General Comments:

=============

1. There is still a typo in the title. "manufacture" should be "manufacturer". I would suggest a shorter title such as:

"Recovery of the 3-dimensional wind and sonic temperature data from a physically-deformed sonic anemometer"

2. The effect of the deformation on the fluctuations and fluxes should be described. It's clear the mean values were affected, but sonics are primarily used to measure fluxes.

3. Thank you for including the code in MATLAB format (Appendix C). Can the actual MATLAB program itself be included as a supplement to the paper?

4. Throughout the manuscript the ultrasonic signals (sound pulses) are described as "flying" between the transducers. This terminology seems a bit colloquial—would a better term be "transmitted"?

5. One thing that struck me in your equations/description is that the speed of sound is shown to be affected by the wind velocity...it seems like most papers I have looked

at describing sonic anemometers use only a single speed of sound which is a function of temperature, humidity. And then the wind velocity affects the time of travel across transducers, but not the speed of sound itself...I think what you have done is correct, but is there another reference that shows the speed of sound depends on the crosswind value (i.e., $c\_i$ vs $c\_{Ti}$).

6. The drawings/schematics in the manuscript are quite nice/clear, but the english language usage needs to be improved (examples from the first 2 pages are below, but there are more throughout the manuscript).

Specific Comments:

=============

\* p.1, l.25, "had been" should be "was"...

\* p.1, l.25, remove, "To recover data from this deformed sonic"

\* p.1, l.30, replace "to the studies on" with "for"...

\* p.2, l.5, what does "structuring" mean?

\* p.2, l.6, what does it mean by "optimized" angles. Optimized for what?

\* p.2, l.11, this reference to "entropy" seems out of place? Don't see entropy mentioned anywhere else in the manuscript...

\* p.2, l.12, "geometry embedded" should be "geometrical information embedded"..

\* p.2, l.15, remove "any more."

\* p.2, l.16, replace "cannot output" with "no longer outputs"

\* p.2, l.23, remove "at the time"

\* p.2, l.23, remove "to which the anemometer can be shipped back with care."

\* p.2, l.28, replace "site" with "situation"

* p.2, l.36, remove "then"

* p.2, l.38-39, awkward sentence, fix the end of it.

* p.3, l.17, It seems odd to mention the funding in the manuscript?

* p.3, l.21, replace "4-way net radiometer" with "4-component radiometer" (also, not necessary to describe the components, the radiation is not really important to the study, so be as brief as possible in this description.)

* p.4, l.7, "unexpectedly various individually"?

* p.4, l.35, replace, "production of recalibration" with "the calibration".

* p.4, sec 2 (and photo in Fig 2). I don't quite understand..there was a CSAT3B there, but you are not comparing the "deformed" sonic results to it (especially for the fluxes)? The best comparison would be to have the "deformed" sonic mounted side-by-side with a "normal" sonic, and then the post-processing correction of the deformed sonic could be evaluated quite well. Was this never done and/or impossible to do (even after it was recalibrated)?

* p.5, eq 3 and 4: probably don't need eq3?

* p.5, l.25, replace "based" with "depending"

* p.8, eq. 21, this is only true for dry air, correct?

* p.11, l.3, isn't the point of the paper verifying that the recovery works?

* p.11, l.22, what does "bare satisfactory" mean?

* p.16, l.2, "Li-Cor" should be "LI-COR".

* p.16, l.9, "popularly used around the world", should be "used around the world". Considering several of the authors work for Campbell Sci. such subjective word choices should not be used.

---

## Referee Comment (RC2) · T. Foken (Referee) · 6 Aug 2018

The paper describes a useful tool for recovering data from sonic anemometers when the mechanical construction of the anemometer has been slightly deformed. This may be of interest for remote stations without permanent control or following deformation during transport to remote areas, as was the case in the paper.

The basic equations of wind vector and of temperature measurements for sonic anemometers are given in the literature, e.g. for wind by Hanafusa et al. (1982) and

for temperature by Kaimal and Gaynor (1991). Furthermore, overview papers and text-books are available (Kaimal and Finnigan, 1994; Aubinet et al., 1999; Aubinet et al., 2012; Foken, 2017). It is unusual to use sensor documentation as a reference (p.2, line 2, p.4, line 32, p. 6 line 19). It is acceptable to use the symbols of these documentations. Furthermore you should highlight (e.g. p. 2, line 12, p. 6 line 23) that the firmware must be known in order to apply your method. This is true for the Campbell instruments but not for all available sensors on the market (or is p. 14 line 37-39 the relevant statement?). For specific firmware problems of Campbell instruments you should check the relevance of the paper by Burns et al. (2012) for your method.

The UW-sonic anemometer (University of Washington, Wyngaard and Zhang, 1985; Zhang et al., 1986) looks like the IRGASON, but is not identical. For the transducer shadow effect, first published by Kaimal (1978), you should use the data from the more recent reference (Horst et al., 2015), p. 5 line 29ff.

You used the calculated sonic temperature rather than the wind measurements as a reference for the accuracy of your corrections, because sensors were missing. Besides the difficulty of making a comparison with the 3D wind vector, there is another reason that the temperature is a more sensitive parameter. This can be shown if you add a small error epsilon to all measured traveling times in Eqs. 3 and 17. You can therefore assume that if the sonic temperature for corrected path lengths is within the accuracy limits of the sensors then this should be realized for the wind components as well.

The separation of the results (Section 4 – 7) and the discussion (Section 8) section is always difficult because of the repetitions. On the other hand, information from Section 8 is necessary for a better understanding of the applied methods (perhaps you can make some modifications), e. g. p. 13 line 18ff: different response times of measured and calculated sonic temperature. This is not a simple lag time. A correction of the dynamical error may be necessary.

Please discuss with the editor whether the software (Appendix C) should be published

in the supplement (not as an Appendix) or in a separate software publication, e.g. on the Zenodo server (https://zenodo.org/).

Minor remarks:

Perhaps you could reduce the number of equations by writing the basic equations in a more general form like Eqs. 3, 4, 12, 13 etc.

p. 3, line 33: information about the used radiation shield of the HMP-sensor is necessary (ventilated?) for Section 8.

p. 11, line 10ff: Could you please give temperature differences in the SI-dimension K. In the present form misunderstanding is possible.

p. 12, line 1: The symbol $cT2$ could be misunderstood because $CT2$ is the standard symbol for the temperature structure parameter; perhaps you can find a better symbol.

p.14, line 2-3: I do not understand the sentence "sonic path becomes shorter by some degree". If the geometry of the sonic anemometer changes below $-20°C$, why can you not correct this effect with your software.

p. 16, line 4-5: Energy balance closure is not a good indicator for data quality (Foken et al., 2012). However your result is in the typical range reported in the literature.

p. 22, line 12: Buck (1981) is not an acceptable reference, because the temperature scale has been changed (ITS-90). A relevant reference is WMO (2014 (update 2017)) or the original reference (Sonntag, 1990).

References:

Aubinet, M., Grelle, A., Ibrom, A., Rannik, Ü., Moncrieff, J., Foken, T., Kowalski, A. S., Martin, P. H., Berbigier, P., Bernhofer, C., Clement, R., Elbers, J., Granier, A., Grünwald, T., Morgenstern, K., Pilegaard, K., Rebmann, C., Snijders, W., Valentini, R., and Vesala, T.: Estimates of the annual net carbon and water exchange of forests: The EUROFLUX methodology, Adv. Ecol. Res., 30, 113-175, 10.1016/S0065-2504(08)60018-

5, 1999.

Aubinet, M., Vesala, T., and Papale, D.: Eddy Covariance: A Practical Guide to Measurement and Data Analysis, Springer, Dordrecht, Heidelberg, London, New York, 438 pp., 2012.

Burns, S. P., Horst, T. W., Jacobsen, L., Blanken, P. D., and Monson, R. K.: Using sonic anemometer temperature to measure sensible heat flux in strong winds, Atmos. Meas. Techn., 5, 2095-2111, 10.5194/amt-5-2095-2012, 2012.

Foken, T., Leuning, R., Oncley, S. P., Mauder, M., and Aubinet, M.: Corrections and data quality in: Eddy Covariance: A Practical Guide to Measurement and Data Analysis, edited by: Aubinet, M., Vesala, T., and Papale, D., Springer, Dordrecht, Heidelberg, London, New York, 85-131, 2012.

Foken, T.: Micrometeorology, 2nd ed., Springer, Berlin, Heidelberg, 362 pp., 2017.

Hanafusa, T., Fujitana, T., Kobori, Y., and Mitsuta, Y.: A new type sonic anemometer-thermometer for field operation, Papers Meteorol. Geophys., 33, 1-19, 1982.

Horst, T. W., Semmer, S. R., and Maclean, G.: Correction of a non-orthogonal, three-component sonic aanemometer for flow Distortion by transducer shadowing, Boundary-Layer Meteorol., 155, 371-395, 10.1007/s10546-015-0010-3, 2015.

Kaimal, J. C.: Sonic Anemometer Measurement of Atmospheric Turbulence, in: Proceedings of the Dynamic Flow Conference 1978 on Dynamic Measurements in Unsteady Flows, edited by: Hansen, B. W., Springer Netherlands, 551-565, 1978.

Kaimal, J. C., and Gaynor, J. E.: Another look to sonic thermometry, Boundary-Layer Meteorol., 56, 401-410, 1991.

Kaimal, J. C., and Finnigan, J. J.: Atmospheric Boundary Layer Flows: Their Structure and Measurement, Oxford University Press, New York, NY, 289 pp., 1994.

Sonntag, D.: Important new values of the physical constants of 1986, vapour pressure

formulations based on the ITC-90, and psychrometer formulae, Z. Meteorol., 40, 340-344, 1990.

WMO: Guide to meteorological instruments and methods of observation, WMO-No. 8, World Meteorological Organization, Geneva, 8th edition, 1128 pp., 2014 (update 2017).

Wyngaard, J. C., and Zhang, S.-F.: Transducer-Shadow Effects on Turbulence Spectra Measured by Sonic Anemometers, J. Atm. Oceanic Techn., 2, 548-558, 10.1175/1520-0426(1985)002<0548:TSEOTS>2.0.CO;2, 1985.

Zhang, S. F., Wyngaard, J. C., Businger, J. A., and Oncley, S. P.: Response characteristics of the U.W. sonic anemometer, J. Atm. Oceanic Techn., 2, 548-558, 1986.
* * *

---

## Author Comment (AC1) · 17 Aug 2018

*Responses to reviewers' comments*

We appreciate very much the constructive and helpful comments from the reviewer. The detailed responses are listed one by one as following:

***Reviewer #1:***
***Major comments***
1. *Examination of the fluctuations in wind velocities and sonic temperature and flux quantities that were influenced by the geometric deformation of sonic anemometer*

   Response: The fluctuations for each wind speed components and sonic temperature are reflected by variance. The variance values of three component wind velocities and sonic temperature in period of two days were analyzed for or the homogeneity between unrecovered and recovered data. The four F-values for three wind speed components and sonic temperature showed the inhomogeneity in variance between unrecovered and recovered data (P < 0.001), which indicates that the geometrical deformation of sonic anemometer did significantly influence the fluctuations in each of its measured variables.
   (see added Section 8.5).

   Figure 8 was added to show the difference in sensible heat flux, latent heat flux, and $CO_2$ flux between unrecovered and recovered data. The differences in the three fluxes are all statistically significant (e.g. all P-value < 0.005, see Figure 8 and Section 8.5).

   The results from the analyses and Figure 8 were added to Conclusion remarks.

2. *English writing*
   Response: Professional English technical writer, Ms. Linda Worlton-Jones, with Campbell Scientific was administratively assigned to polish the English writing.

***General comments***
1. *Suggestion to shorten the title*
   Response: The title was shortened as suggested to:

   "Recovery of the 3-dimensional wind and sonic temperature data from a physically deformed sonic anemometer"

2. *Effect of the deformation on the fluctuations and fluxes*
   Response: See response to major comment 1.

3. *Actual MATLAB program*
   Response: The program in Appendix C is the actual one, but it excludes the code lines for dialogue interface. The other referee, Dr. Thomas Foken, suggested that this section should be published in a separate publication. He also advised us to seek an opinion from the Editor. The Editor (Dr. Laura Bianco) agreed with Dr. Foken's suggestion. We will work on this program in a publication shape. At this

stage, we would keep Appendix C as is. It is noted in Appendix C that the operational code now can be requested from corresponding authors.

4. *Terminology: Flying and transmitted.*
   Response: "Transmitted" is right in terminology although we often use "flying" for our training seminars and in-house communications. The term of "Flying" was replaced with "transmitted".

5. *Crosswind effect*
   Response: The crosswind effect on measurements of speed of sound is corrected inside the operating system of sonic anemometer. The speed of sound from each of three sonic paths is separately corrected and the three corrected speeds are used to estimate the sonic temperature. The reference of Schotanus et al. (1983) was added as citation. This reference shows how crosswind influences the measurements of speed of sound [see Figure 1 and equations (1) and (2) in Schotanus et al. (1983)].

   Schotanus, P., F. T. M. Nieuwstadt, and H. A. R. de Bruin. 1983. Temperature measurement with a sonic anemometer and its application to heat and moisture fluxes, Boundary-Layer Meteorology 26: 81-93.

6. *Drawings/schematics and English*
   Response: Thank you so much for your positive comments on the drawings/schematics in the manuscript and specific comments for the revisions of English.

   * p.1, l.25, "had been" should be "was"...
   Response: Revised as suggested.

   * p.1, l.25, remove, "To recover data from this deformed sonic"
   Response: Removed as suggested.

   * p.1, l.30, replace "to the studies on" with "for"...
   Response: Replaced as suggested.

   * p.2, l.5, what does "structuring" mean?
   Response: Means "forming" three paths in a designed geometry in structure. For simplicity, "structuring" was replaced with "forming" and the whole sentence was revised.

   * p.2, l.6, what does it mean by "optimized" angles. Optimized for what?
   Response: For wind measurements. The sentence was revised as:
   "The three paths are situated as optimized angles for wind measurements in the 3D anemometer coordinate system, …….".

   *p.2,l.11,this reference to "entropy" seems out of place? Don't see entropy mentioned anywhere else in the manuscript...
   Response: The term of "entropy" was replaced with "heat property".

   * p.2, l.12, "geometry embedded" should be "geometrical information embedded".
   Response: Revised as suggested.

\* p.2, l.15, remove "any more."
Response: Removed as suggested.

\* p.2, l.16, replace "cannot output" with "no longer outputs"
Response: Replaced as suggested.

\* p.2, l.23, remove "at the time"
Response: Removed as suggested.
\* p.2, l.23, remove "to which the anemometer can be shipped back with care."
Response: We would like to keep this writing. If the anemometer was shipped back as usual without care, it might be deformed again in transportation. If deformed again, its geometry re-measurements after back to manufacturer would not be representative to sonic geometry during field measurements, which would bring uncertainties to the data recovery.

\* p.2, l.28, replace "site" with "situation"
Response: "In such a site" was revised as "From such a site".

\* p.2, l.36, remove "then"
Response: Removed as suggested.

\* p.2, l.38-39, awkward sentence, fix the end of it.
Response: Fixed as "More importantly, the 2015 data was also needed by related projects for collaborations."

\* p.3, l.17, It seems odd to mention the funding in the manuscript?
Response: Removed the wording related to the funding.

\* p.3, l.21, replace "4-way net radiometer" with "4-component radiometer" (also, not necessary to describe the components, the radiation is not really important to the study, so be as brief as possible in this description.)
Response: Revised as suggested and removed the words how net radiation is measured.

\* p.4, l.7, "unexpectedly various individually"?
Response: Revised the sentence as
….. that the sonic temperature values from the three sonic paths unexpectedly deviated around -12, 5, and -7 ℃ …….

\* p.4, l.35, replace "production of recalibration" with "the calibration".
Response: The path lengths are measured in two cases: production calibration and return calibration processes. The phrase of "production calibration and return recalibration" may be wordy. The phrase of "during production or recalibration" is to express our description.

\* p.4, sec 2 (and photo in Fig 2). I don't quite understand there was a CSAT3B there, but you are not comparing the "deformed" sonic results to it (especially for the fluxes)? The best comparison would be to have the "deformed" sonic mounted side-by-side with a "normal" sonic, and then the post-processing correction of the

deformed sonic could be evaluated quite well. Was this never done and/or impossible to do (even after it was recalibrated)?

Response: The photo was taken after the deformed IRGASON was replaced with the manufacture-provided swap unit. Before the deformed IRGASON was thoroughly inspected and checked by the manufacturer, we were not 100% sure what caused the incorrect measurements of sonic temperature. What we were worried about was that IRGASON could not be used in such cold conditions. To ensure the sonic temperature data, a CSAT3B was installed as an alternative although the swap unit was installed. The deformed IRGASON and CSAT3B were not deployed side-by-side. For this case, the best comparison as you suggested is impossible.

* p.5, eq 3 and 4: probably don't need eq3?
Response: In Figure 1, we must use a specific sonic path to illustrate the measurements of wind speed and speed of sound. For a better spatial illustration, the third sonic path was used. As a result, equations (1) and (2) are particularly referred to the sonic path and equation (3) is used to make transition from the third sonic path to the $i$th sonic path where i = 1, 2, or 3. We feel that the use of equation (3) could make an entrance-level reader easier.

* p.5, l.25, replace "based" with "depending"
Response: Replaced as suggested.

* p.8, eq. 21, this is only true for dry air, correct?
Response: We cannot correctly answer this question simply using either "correct" or "incorrect". This question would be better answered using the following explanations.

In acoustics, the speed of sound ($c$) in a homogeneous gaseous medium as in the atmospheric surface-layer flows is well defined as (Barrett, E.W., V.E. Suomi. 1949. Preliminary report on temperature measurement by sonic means. Journal of Meteorology 6: 273-276)

$$c^2 = \gamma \frac{P}{\rho} \qquad\qquad (R1)$$

Where $\gamma$ is the ratio of moist air specific heat at constant pressure to moist air specific heat at constant volume, and $\rho$ is moist air density. Substituting the ideal gas equation,

$$P = R_a \rho T \qquad\qquad (R2)$$

where $R_a$ is the gas constant of moist air. Using two equations above, $T$ can be related to $c$ as:

$$T = \frac{c^2}{\gamma R_a} \qquad\qquad (R3)$$

This equation enlightens the use of measured $c$ for $T$ calculation; however, both $\gamma$ and $R_a$ depend on air humidity undermined by any sonic anemometer; equation

(R3) is, therefore, not applicable for $T$ calculations inside a sonic anemometer. Alternatively, $\gamma$ is replaced with its counterpart for dry air [$\gamma_d$ (1.4003), the ratio of dry air specific heat at constant pressure (1,004 J K$^{-1}$ kg$^{-1}$) to dry air specific heat at constant volume (717 J K$^{-1}$ kg$^{-1}$)] and $R_a$ is replaced with its counterpart for dry air ($R_d$, gas constant for dry air, being 287.04 J K$^{-1}$ kg$^{-1}$). After both replacements in equation (R3) and although, in magnitude, $\gamma_d$ is close to $\gamma$ and $R_d$ is close to $R_a$, the variable in its left hand side is not a measure of $T$ anymore. Instead, it is defined as sonic temperature denoted by $T_s$:

$$T_s = \frac{c^2}{\gamma_d R_d} \qquad\qquad (R4)$$

This equation is the equation (21) in manuscript. It is the definition of sonic temperature.

\* p.11, l.3, isn't the point of the paper verifying that the recovery works?
Response: The recovery of wind data does not need verification because the equations (10) to (16) for recovering the wind data do not include any assumption and approximation.

\* p.11, l.22, what does "bare satisfactory" mean?
Response: The phrase of "bare satisfactory" means marginally satisfactory. The word of bare was replaced with "less". The related context ahead of this sentence was revised accordingly.

\* p.16, l.2, "Li-Cor" should be "LI-COR".
Response: Revised as suggested.

\* p.16, l.9, "popularly used around the world", should be "used around the world". Considering several of the authors work for Campbell Sci. such subjective word choices should not be used.
Response: Revised as suggested.

---

## Author Response (AR1)

Sun Yat-Sen University
School of Atmospheric Sciences
Zhuhai 519082, China

August 17, 2018

RE: Responses to reviewers' comments on manuscript amt-2018-92_RC1

Dr. Laura Bianco
Associate Editor
Atmospheric Measurement Techniques

Dear Dr. Laura Bianco,

We have been really appreciated with both reviewers' for their comments which significantly improve the manuscript. In particular, we thank anonymous referee #1 so much for his/her patience with our non-native English writing and also thank Dr. Foken so much for his advices in use of literature. Following their comments and advices, we revised our manuscript and addressed their comments in the revision. Please find our detailed responses to the reviewers' comments below as well as a description of how the manuscript has been improved.

With best regards,

Qinghua Yang
On behalf of the co-authors

**Anonymous Referee**

**Major comments**

1. *Examination of the fluctuations in wind velocities and sonic temperature and flux quantities that were influenced by the geometric deformation of sonic anemometer*

   Response: The fluctuations for each wind speed components and sonic temperature are reflected by variance. The variance values of three component wind velocities and sonic temperature in period of two days were analyzed for or the homogeneity between unrecovered and recovered data. The four F-values for three wind speed components and sonic temperature showed the inhomogeneity in variance between unrecovered and recovered data ($P < 0.001$), which indicates that the geometrical deformation of sonic anemometer did significantly influence the fluctuations in each of its measured variables. (see added Section 8.5).

   Figure 8 was added to show the difference in sensible heat flux, latent heat flux, and $CO_2$ flux between unrecovered and recovered data. The differences in the three fluxes are all statistically significant (e.g. all P-value $< 0.005$, see Figure 8 and Section 8.5).

   The results from the analyses and Figure 8 were added to Conclusion remarks.

2. *English writing*

Response: Professional English technical writer, Ms. Linda Worlton-Jones, with Campbell Scientific was administratively assigned to polish the English writing.

**General comments**

1. *Suggestion to shorten the title*
   Response: The title was shortened as suggested to:

   "Recovery of the 3-dimensional wind and sonic temperature data from a physically deformed sonic anemometer"

2. *Effect of the deformation on the fluctuations and fluxes*
   Response: See response to major comment 1.

3. *Actual MATLAB program*
   Response: The program in Appendix C is the actual one, but it excludes the code lines for dialogue interface. The other referee, Dr. Thomas Foken, suggested that this section should be published in a separate publication. He also advised us to seek an opinion from the Editor. The Editor (Dr. Laura Bianco) agreed with Dr. Foken's suggestion. We will work on this program in a publication shape. At this stage, we would keep Appendix C as is. It is noted in Appendix C that the operational code now can be requested from corresponding authors.

4. *Terminology: Flying and transmitted.*
   Response: "Transmitted" is right in terminology although we often use "flying" for our training seminars and in-house communications. The term of "Flying" was replaced with "transmitted".

5. *Crosswind effect*
   Response: The crosswind effect on measurements of speed of sound is corrected inside the operating system of sonic anemometer. The speed of sound from each of three sonic paths is separately corrected and the three corrected speeds are used to estimate the sonic temperature. The reference of Schotanus et al. (1983) was added as citation. This reference shows how crosswind influences the measurements of speed of sound [see Figure 1 and equations (1) and (2) in Schotanus et al. (1983)].

   Schotanus, P., F. T. M. Nieuwstadt, and H. A. R. de Bruin. 1983. Temperature measurement with a sonic anemometer and its application to heat and moisture fluxes, Boundary-Layer Meteorology 26: 81-93.

6. *Drawings/schematics and English*
   Response: Thank you so much for your positive comments on the drawings/schematics in the manuscript and specific comments for the revisions of English.

   * p.1, l.25, "had been" should be "was"...
   Revised as suggested.

   * p.1, l.25, remove, "To recover data from this deformed sonic"
   Removed as suggested.

   * p.1, l.30, replace "to the studies on" with "for"...
   Replaced as suggested.

\* p.2, l.5, what does "structuring" mean?

Means "forming" three paths in a designed geometry in structure. For simplicity, "structuring" was replaced with "forming" and the whole sentence was revised.

\* p.2, l.6, what does it mean by "optimized" angles. Optimized for what?

For wind measurements. The sentence was revised as:

"The three paths are situated as optimized angles for wind measurements in the 3D anemometer coordinate system, ......."

\*p.2,l.11,this reference to "entropy" seems out of place? Don't see entropy mentioned anywhere else in the manuscript...

The term of "entropy" was replaced with "heat property"

\* p.2, l.12, "geometry embedded" should be "geometrical information embedded"..

Revised as suggested.

\* p.2, l.15, remove "any more."

Removed as suggested.

\* p.2, l.16, replace "cannot output" with "no longer outputs"

Replaced as suggested.

\* p.2, l.23, remove "at the time"

Removed as suggested.

\* p.2, l.23, remove "to which the anemometer can be shipped back with care."

We would like to keep this writing. If the anemometer was shipped back as usual without care, it might be deformed again in transportation. If deformed again, its geometry re-measurements after back to manufacturer would not be representative to sonic geometry during field measurements, which would bring uncertainties to the data recovery.

\* p.2, l.28, replace "site" with "situation"

"In such a site" was revised as "From such a site".

\* p.2, l.36, remove "then"

Removed as suggested.

\* p.2, l.38-39, awkward sentence, fix the end of it.

Fixed as "More importantly, the 2015 data was also needed by related projects for collaborations."

\* p.3, l.17, It seems odd to mention the funding in the manuscript?

Removed the wording related to the funding.

\* p.3, l.21, replace "4-way net radiometer" with "4-component radiometer" (also, not necessary to describe the components, the radiation is not really important to the study, so be as brief as possible in this description.)

Revised as suggested and removed the words how net radiation is measured.

\* p.4, l.7, "unexpectedly various individually"?

Revised the sentence as

….. that the sonic temperature values from the three sonic paths unexpectedly deviated around -12, 5, and -7 ℃ …….

\* p.4, l.35, replace "production of recalibration" with "the calibration".
The path lengths are measured in two cases: production calibration and return calibration processes. The phrase of "production calibration and return recalibration" may be wordy. The phrase of "during production or recalibration" is to express our description.

\* p.4, sec 2 (and photo in Fig 2). I don't quite understand there was a CSAT3B there, but you are not comparing the "deformed" sonic results to it (especially for the fluxes)? The best comparison would be to have the "deformed" sonic mounted side-by-side with a "normal" sonic, and then the post-processing correction of the deformed sonic could be evaluated quite well. Was this never done and/or impossible to do (even after it was recalibrated)?

The photo was taken after the deformed IRGASON was replaced with the manufacture-provided swap unit. Before the deformed IRGASON was thoroughly inspected and checked by the manufacturer, we were not 100% sure what caused the incorrect measurements of sonic temperature. What we were worried about was that IRGASON could not be used in such cold conditions. To ensure the sonic temperature data, a CSAT3B was installed as an alternative although the swap unit was installed. The deformed IRGASON and CSAT3B were not deployed side-by-side. For this case, the best comparison as you suggested is impossible.

\* p.5, eq 3 and 4: probably don't need eq3?
In Figure 1, we must use a specific sonic path to illustrate the measurements of wind speed and speed of sound. For a better spatial illustration, the third sonic path was used. As a result, equations (1) and (2) are particularly referred to the sonic path and equation (3) is used to make transition from the third sonic path to the $i$th sonic path where i = 1, 2, or 3. We feel that the use of equation (3) could make an entrance-level reader easier.

\* p.5, l.25, replace "based" with "depending"
Replaced as suggested.

\* p.8, eq. 21, this is only true for dry air, correct?
We cannot correctly answer this question simply using either "correct" or "incorrect". This question would be better answered using the following explanations.
  In acoustics, the speed of sound ($c$) in a homogeneous gaseous medium as in the atmospheric surface-layer flows is well defined as (Barrett, E.W., V.E. Suomi. 1949. Preliminary report on temperature measurement by sonic means. Journal of Meteorology 6: 273-276)

$$c^2 = \gamma \frac{P}{\rho} \qquad\qquad \text{(R1)}$$

Where $\gamma$ is the ratio of moist air specific heat at constant pressure to moist air specific heat at constant volume, and $\rho$ is moist air density. Substituting the ideal gas equation,

$$P = R_a \rho T \qquad\qquad \text{(R2)}$$

where $R_a$ is the gas constant of moist air. Using two equations above, $T$ can be related to $c$ as:

$$T = \frac{c^2}{\gamma R_a} \qquad \text{(R3)}$$

This equation enlightens the use of measured $c$ for $T$ calculation; however, both $\gamma$ and $R_a$ depend on air humidity undermined by any sonic anemometer; equation (R3) is, therefore, not applicable for $T$ calculations inside a sonic anemometer. Alternatively, $\gamma$ is replaced with its counterpart for dry air [$\gamma_d$ (1.4003), the ratio of dry air specific heat at constant pressure (1,004 J K$^{-1}$ kg$^{-1}$) to dry air specific heat at constant volume (717 J K$^{-1}$ kg$^{-1}$)] and $R_a$ is replaced with its counterpart for dry air ($R_d$, gas constant for dry air, being 287.04 J K$^{-1}$ kg$^{-1}$). After both replacements in equation (R3) and although, in magnitude, $\gamma_d$ is close to $\gamma$ and $R_d$ is close to $R_a$, the variable in its left hand side is not a measure of $T$ anymore. Instead, it is defined as sonic temperature denoted by $T_s$:

$$T_s = \frac{c^2}{\gamma_d R_d} \qquad \text{(R4)}$$

This equation is the equation (21) in manuscript. It is the definition of sonic temperature.

\* p.11, l.3, isn't the point of the paper verifying that the recovery works?
The recovery of wind data does not need verification because the equations (10) to (16) for recovering the wind data do not include any assumption and approximation.

\* p.11, l.22, what does "bare satisfactory" mean?
The phrase of "bare satisfactory" means marginally satisfactory. The word of bare was replaced with "less". The related context ahead of this sentence was revised accordingly.

\* p.16, l.2, "Li-Cor" should be "LI-COR".
Revised as suggested.

\* p.16, l.9, "popularly used around the world", should be "used around the world". Considering several of the authors work for Campbell Sci. such subjective word choices should not be used.
Revised as suggested

**Dr. Thomas Foken**

*Major comments*

1. *Applications*

Response: Thank you for the positive comments on the applications of this study. The equations and algorithms are useful to recover data not only from geometrically deformed sonic anemometers, but also from CSAT3 sonic anemometers using unmatched electronic boxes in the field. The geometry data of each CSAT3 sonic anemometer embedded into its electronic box. If a CSAT3 head is used with an electronic box for other CSAT3, this CSAT3 head would use wrong geometry data to calculate the 3D wind and sonic temperature. Such cases are equivalent to the data acquisition using a sonic anemometer slightly deformed. The measured data could be recovered using the geometry data from the unmatched electronic box for other CSAT3 (equivalent to geometry data before

deformation in this study) and from its own electronic box (i.e. geometry data after deformation in this study).

The geometry data can be requested from manufacturer. Over years, the requests to recover the data from such cases were received, but the equations and algorithms to recover the sonic temperature data with full satisfaction were not available because sonic temperature was corrected for the crosswind effect inside the sonic anemometer OS separately for each of three sonic paths, which complicates the recovery of sonic temperature. This study greatly improved the recovery of sonic temperature data from slightly deformed sonic anemometers and CSAT3 sonic anemometers using unmatched electronic box. The newer models of sonic anemometers such as CSAT3B, CSAT3A, and IRGASON sonic anemometer embed the geometry data inside a component of anemometer head (e.g. an electronic chip attached to sonic anemometer head connector); therefore, considering the length of manuscript, we did not tell such a story.

2. Citations of manufacturer's documents

Response: The citations of some manufacturer's documents were removed or replaced with journal publications. In particular, earliest Hanafusa et al. (1982) and most recent Foken (2017) were added. For sensor specifications, manufacture documentations have to be referenced.

3. Highlight firmware for sonic anemometer

Response: The version number of EC100 OS for sonic anemometer was EC100.04.10 (02/25/2014) when this anemometer was used in the Antarctic. This version number was added in the statement related to EC100 in Section 2.

The equations and algorithms in this study are not relevant to the version number of sonic anemometer OS, but the application of the equations and algorithms needs the embedded geometry data and the embedded transform matrixes inside the sonic anemometer firmware. The geometry data and transform matrixes are unique for each Campbell sonic anemometer and are identified by serial number. These data and matrixes for sonic anemometer SN: 1131 in this study were acquired from Campbell Scientific and were given in Table A1 and matrixes (A3) to (A6) in Appendix A. Following Dr. Foken's advice, the information related these data and matrix was highlighted in related lines as pointed by Dr. Foken and other related statements.

Additionally, Burns et al. (2012, including Larry Jacobsen) found the underestimation in sonic temperature fluctuations when wind speed > 8 m/s if CSAT3 OS 4.0 was used. Larry Jacobsen fixed the problem encountered in this particular version of CSAT3 OS 4.0.

Burns, S.P., Horst, T.W., Jacobsen, L., Blanken, P.D., Monson, R. K. 2012. Using sonic anemometer temperature to measure sensible heat flux in strong winds, Atmos. Meas. Techn., 5, 2095-2111.

4. Transducer-shadow correction

Response: After Horst et al. (2015), Larry Jacobsen implemented transducer-shadow correction into CSAT3A, IRGASON sonic anemometer, and CSAT3B as an option. For CSAT3A and IRGASON, OS EC100.07.01 or later has this option. If this option is used,

the data recovery must use the equations and algorithms including shadow correction.

The parameters in the correction equation are not same as those used by Wyngaard and Zhang (1985). Using the sonic transducer diameter of 0.6 cm and ratio of sonic path length to diameter (19.25), the parameters were determined based the Figure 5, equation (1a), and Table 1 in Wyngaard and Zhang (1985) as indicated by equation (7) in our manuscript. The equations related to transducer-shadow correction are the same as those used inside IRGASON, OS EC100.07.01 after Horst et al. (2015); therefore, the citation for equation (7) was revised as "Following Host et al. (2015) based on Wyngaard and Zhang (1985), ………….."

Horst, T.W., Semmer, S.R., Maclean, G. 2015. Correction of a non-orthogonal, three-component sonic anemometer for flow Distortion by transducer shadowing, Boundary-Layer Meteorol., 155, 371-395.

Wyngaard, J.C., Zhang, S.F. 1985. Transducer-shadow effects on turbulence spectra measured by sonic anemometers, J. Atm. Oceanic Techn. 2: 548-558,

5.  More sensitivity of sonic temperature to a measurement error
Response: We are really appreciated with your deep and substantial insight about the issue of verification on the data recovery. From equations (3), (17), and (21), the error analysis can be derived. Sonic temperature is sensitive at one order higher than wind speed to the errors in measurements of sonic path lengths and ultrasonic signal travel times; therefore, the calculated sonic temperature instead of wind speed was used to verify the data recovery in Section 8. Your suggested argument "…..if the sonic temperature for corrected path lengths is within the accuracy limits of the sensors then this should be realized for the wind components as well.", however, consider the length of our manuscript, we do not prefer to add more equations in our manuscript for error analysis mentioned in this response. Instead, this comment was cited in our discussion section.

6.  Different response time of sensors

Response: For the data mean of half hour, the response time is not an issue. For simplicity, the discussion related to the time lag was removed.

7.  Discuss with the editor about the software (Appendix C) publication
Response: We have discussed with the Editor (Dr. Laura Bianco), the Editor agreed with your suggestion. We will work on this program in a publication shape. At this stage, we would keep Appendix C as is. It is noted in Appendix C that the operational code now can be requested from corresponding authors.

*General comments*
Perhaps you could reduce the number of equations by writing the basic equations in a more general form like Eqs. 3, 4, 12, 13 etc.
Response: In Figure 1, we should use a specific sonic path to illustrate the measurements of wind speed and speed of sound. For a better spatial illustration, the third sonic path was used. As a result, equations (1) and (2) are particularly referred to the sonic path and equation (3) is

used to make transition from the third sonic path to the $i$th sonic path where i = 1, 2, or 3. We feel that the use of equation (3) can make an entrance-level reader easier. Same to other equations.

p. 3, line 33: information about the used radiation shield of the HMP-sensor is necessary (ventilated?) for Section 8.
Response: It was not fan-ventilated. It was naturally ventilated. Power is a limited factor in the Antarctic area.

p. 11, line 10ff: Could you please give temperature differences in the SI-dimension K. In the present form misunderstanding is possible.
Response: The unit of degree C for temperature differences was revised as K. Throughout the manuscript and figures, K is used for the unit for temperature difference.

p. 12, line 1: The symbol cT2 could be misunderstood because CT2 is the standard symbol for the temperature structure parameter; perhaps you can find a better symbol.
Response: All $c_{T1}$, $c_{T2}$, $c_{T3}$, and $c_{Ti}$ are renamed as $c_{01}$, $c_{02}$, $c_{03}$, and $c_{0i}$ where subscript 0 indicates the speed of sound at the crosswind speed equal to 0. This revision was made through the manuscript and figures.

p.14, line 2-3: I do not understand the sentence "sonic path becomes shorter by some degree". If the geometry of the sonic anemometer changes below–20∘C, why can you not correct this effect with your software.
Response: Thermo-expansion or -contraction happens to the whole body of sonic anemometer structure. As a result, the sonic path can be longer or shorter, which can influence the measurement. However, this topic goes beyond the scope of this study that recovers the data from geometrically deformed sonic anemometer to those from a normal one.

p. 16, line 4-5: Energy balance closure is not a good indicator for data quality (Foken et al., 2012). However your result is in the typical range reported in the literature.
Response: Yes. Following Foken et al., (2012), the discussion was revised.

p. 22, line 12: Buck (1981) is not an acceptable reference, because the temperature scale has been changed (ITS-90). A relevant reference is WMO (2014 (update 2017)) or the original reference (Sonntag, 1990).
Response: Thank you so much for your update. LI-COR and Campbell Scientific, the two manufacturers to manufacture $H_2O$-related gas analyzers for flux measurements, have been using Buck (1981) for their calculations and $H_2O$ span. Campbell Scientific will accept your recommendation to switch Buck (1981) to Sonntag (1990) for future use. For this study now, the use of Buck (1981) is consistent with the same use by LI-COR and Campbell Scientific.

[revised manuscript text omitted]
 $\cancel{c_{Ti}^2}\,c_{0i}^2$. The average of $\cancel{c_{T1}^2, c_{T2}^2, \text{and } c_{T3}^2}\,c_{01}^2, c_{02}^2, \text{and } c_{03}^2$ can be calculated from Eq. (22) because $T_s$ is an output variable of sonic anemometer. Without a measurement error and random error, the three $\cancel{c_{Ti}}c_{0i}$ should be the same independent of flow speed because they are the true speed of sound instead of measured speed of sound along an individual sonic path (Schotanus et al., 1983; Liu et al., 2001); Therefore, $\cancel{c_{Ti}^2}\,c_{0i}^2$ can be reasonably approximated using the average of three $\cancel{c_{Ti}^2}\,c_{0i}^2$ as $\cancel{c_T^2}\,c_0^2$, given by:

$$\cancel{c_{ci}^2 - c_i^2 = \left(c_T^2 - U_T^2 + u_{Ti}^2\right)\frac{\Delta d_{Ti}^2}{d_i^2}} \quad c_{ci}^2 - c_i^2 = \left(c_0^2 - U_T^2 + u_{Ti}^2\right)\frac{\Delta d_{Ti}^2}{d_i^2}$$

(31)

where $\cancel{c_T^2}\ c_0^2$ can be computed from Eq. (22) as.

$$\cancel{c_T^2 = \gamma_d R_d \left(T_s + 273.15\right)}\ c_0^2 = \gamma_d R_d \left(T_s + 273.15\right)$$
(32)

[revised manuscript text omitted]

---

## Author Response (AR2)

Sun Yat-Sen University
School of Atmospheric Sciences
Zhuhai 519082, China

September 23, 2018

RE: Responses to reviewer's comments on manuscript amt-2018-92_RC2

Dr. Laura Bianco
Associate Editor
Atmospheric Measurement Techniques

Dear Dr. Bianco,

Thank you so much for your favorable consideration of our manuscript publication at this stage of review and revision. We have been really appreciated with both reviewers' inputs in the review processes. In particular, we have felt lucky to have the reviews of well-known scientist, Dr. Foken, on our work. Both reviewers have been appreciated by us in the section of Acknowledgement. In response to referee #1, we revised our manuscript to address his/her comments below.

Additionally, to minimize the errors throughout the manuscript, Dr. Zhou, Dr. Zheng, and I thoroughly reviewed the whole manuscript, again. To keep consistent in expressions and presentations throughout the manuscript, we edited several words and several sentences as well as updated reference list.

Again, thank you so much for your favorable consideration.

Sincerely,

Qinghua Yang, Ph.D.
Professor on Atmospheric Sciences

**Response to Referee #1**

1.    Following up on a comment by Dr. Foken. It seems that the firmware version of sonic anemometer that was used in the study should be clearly stated. I also disagree that Larry Jacobsen "fixed the problem" with CSAT3 OS 4.0. An empirical solution to the problem was determined, but the measured transit times with a CSAT3 using OS Version 4 are incorrect (in high winds). It's not clear to me that the correction method proposed here will work with CSAT3s using OS ver4 and this should be discussed/clarified.

**Response**: About Dr. Foken's comment on CSAT3 OS version 4, we had different interpretation from referee #1. Our interpretation was to indicate the sonic anemometer OS version (i.e. EC100 OS version for IRGASON anemometer, see line 31 on page 3) used by this physically deformed IRGASON sonic anemometer.

CSAT3 OS version 4 was an old defect version and was no longer to be used. Because CSAT3 using OS version 4 could not output the correct data, the equations and algorithms must not work for the data measured by a CSAT3 anemometer using OS version 4. In light of this comment, this issue was further clarified in discussion (see lines 13 and 14 on page 16).

2.    Related to some comments in section 8.5: How far can a sonic be deformed and this method will still work? Do the authors have any idea about this? It might be worth some clarification if this method is only possible for relatively small deformations.

**Response**: A geometrically deformed sonic anemometer outputs erroneous data. These data may be recoverable or unrecoverable, depending on the degree of deformation. The geometric deformation of sonic anemometer is in three dimensions. We are not able to evaluate how far the geometric deformation of sonic anemometer in three dimensions could cause the data unrecoverable, but we do have an idea about this issue.

If the degree is too large, the sonic anemometer cannot perform its normal measurements for transmitting time of ultrasonic signals along every sonic path. In this case, a Campbell sonic anemometer sets high for one to six of its first six measurement warning flags [low amplitude, high amplitude, poor signal lock, large sonic temperature difference, ultrasonic signal loss, and calibration signature error. See Table 10-2 in Campbell Scientific Inc. (2018)]. The geometrical deformation in sonic paths could trigger one or two flags high that indicate poor signal lock and/or ultrasonic signal loss. Anyway, in case that any of the six warning flags from a deformed sonic anemometer was frequently, regularly, or continuously high, the erroneous data must not be recoverable. While all six warning flags are low under normal running conditions, the transmitting time of ultrasonic signals along each sonic path is correctly measured and the data should be recoverable. The 3D wind data can be recovered without uncertainty although there is little uncertainty in sonic temperature [see Eqs (33) and (40)]. The discussion about this issue was added to Section 8.5.

*Campbell Scientific Inc.: IRGASON Integrated $CO_2/H_2O$ Open-Path Gas Analyzer and 3D Sonic Anemometer, Instruction Manual, pp 63, Logan, UT, 2018.*

3.    Re: MATLAB program, I don't understand what is meant by this program should be

included in a "separate publication"? The EGU journals have the ability to include a "Supplement" with published papers...as I understand it, it would be easy and appropriate to include the actual program in the Supplement. If you want to leave the listing in Appendix C, that is fine, but please also include the actual program in the supplement as well.

**Response**: Dr. Foken suggested us to consult with the Editor to publish the code in a separate publication and she agreed his suggestion over an email on Aug 10. So far, we have not found an appropriate avenue to publish it. This code is simple and short although it is little long for an appendix. It would be easier to readers if it is published in the appendix to this manuscript. To fit to an appendix, this code was revised as an executable code if compiled in MATLAB. The electronic version of this code is available from the corresponding authors, which is noted in the beginning of code (see Appendix C).

Additionally, the code was reviewed again. The notation of $c_{Ti}$ was replaced with $c_{0i}$ in consistent with those in the equations inside this manuscript.

4.   It seems like there was a mistake in the Interactive Discussion where the author replies, "AC1" and "AC3" are exactly the same file? ie,

295906 Aug 20 12:38 amt-2018-92-AC1-supplement.pdf
295906 Aug 20 12:37 amt-2018-92-AC3-supplement.pdf

Perhaps there was a mistake in choosing the file when the authors uploaded in the reply?

**Response**: This was caused by my wrong operation during loading files, which was realized immediately after loading. To remove the wrong upload, I contacted with Editorial Support, Ms. Anna Wenzel, over an email on Aug 21. Her reply is given below:

*Dear Qinghua,*
*Thank you very much for your email.*
*Since all interactive comments are fully citable, we should not simply delete them. For this reason, I kindly ask you to for your understanding that we cannot remove AC1 from the discussion.*
*Kind regards,*
Anna

.

[revised manuscript text omitted]